# Merging TEMPEST Microwave and GOES-16 Geostationary IR soundings for improved water vapor profiles

Chia-Pang Kuo[1], Christian Kummerow[1]

[1]Department of Atmospheric Science, Colorado State University, Fort Collins, CO 80523, USA

Correspondence to: Chia-Pang Kuo (chia-pang.kuo@colostate.edu)

**Abstract.** The Temporal Experiment for Storms and Tropical Systems Demonstration (TEMPEST-D) demonstrated the capability of CubeSat satellites to provide high-quality, stable microwave signals for estimating water vapor, clouds, and precipitation from space. Unlike the operational NOAA and MetOp series satellites, which combine microwave and hyperspectral infrared sensors on the same platforms to optimize retrievals, CubeSat radiometers such as TEMPEST do not carry additional sensors. In such cases, the high temporal and spatial resolution and multi-channel measurements from the Advanced Baseline Imager (ABI) on the next-generation series of Geostationary Operational Environmental Satellites (GOES-R) are ideal for assisting these smaller, stand-alone radiometers. Based on sensitivity tests, the water vapor retrievals from TEMPEST are improved by adding water-vapor-sounding, window and $CO_2$ channels at 6.2, 6.9, 7.3, 8.4, 10.3, 11.2, 12.3 and 13.3 μm from ABI, which help to increase the vertical resolution of soundings and reduce retrieval errors. Adding three ABI water-vapor-sounding channels, under clear sky conditions, retrieval biases and root-mean-square errors improve by approximately 10 %, while under cloudy skies, biases remain unchanged, but root-mean-square errors still decrease by 5 %; meanwhile, retrieval biases and root-mean-square errors are substantially reduced by adding more information from eight ABI bands in both clear and cloudy skies. Humidity soundings are also validated using coastal radiosonde data from the Integrated Global Radiosonde Archive (IGRA) from 2019 to 2020. When ABI indicates clear skies, water vapor retrievals improve somewhat by decreasing the overall bias in the microwave only estimate by roughly 10 %, although layer root-mean-square errors remain roughly unchanged at 1 g/kg when three or eight ABI channels are added. When ABI indicates cloudy conditions, there is little change in the results. The small number of matched radiosondes may limit the observed improvement.

## 1.    Introduction

The Temporal Experiment for Storms and Tropical Systems Demonstration (TEMPEST-D; Reising et al., 2018) mission was designed to demonstrate the capability of a small radiometer on board a 6U CubeSat satellite for deriving clouds, water vapor, and precipitation. The CubeSat, including the flight system and the TEMPEST-D radiometer, is 10 cm x 20 cm x 34 cm and weighs 11.2 kg. Although the size of the TEMPEST-D is much smaller than instruments such as the operational Microwave Humidity Sounders (MHS on NOAA-18/19 and MetOp-A/B/C), which weigh about 63 kg, the TEMPEST-D radiometer demonstrated the capability to provide comparable well-calibrated microwave (MW) measurements (Berg et al., 2021; Brown et al., 2023). In addition, Schulte et al. (2020) introduced the bias correction of Earth incidence angle (EIA) (Schulte and Kummerow, 2019) in the Optimal Estimation (OE; Rodgers, 2000) framework with TEMPEST-D and demonstrated the potential of getting consistent retrievals from a fleet of TEMPEST sensors observing the same spot with different EIAs. Radhakrishnan

et al. (2022) estimated surface rainfall by machine-learning methods and showed that retrieved rainfall using TEMPEST-D channels was consistent with the multi-radar/multi-sensor system (MRMS) rainfall products over the Continental United States. The success of TEMPEST-D led to flying a second TEMPEST unit in conjunction with the Compact Ocean Wind Vector Radiometer (COWVR; https://podaac.jpl.nasa.gov/COWVR-TEMPEST) currently in orbit aboard the International Space Station.

Several studies have shown the capability of retrieving surface and atmospheric variables over the ocean under non-raining conditions using Optimal Estimation (OE) techniques. Elsaesser and Kummerow (2008) retrieved total precipitable water (TPW), surface wind, and cloud liquid water path (CLWP) using observations from the Advanced Microwave Scanning Radiometer-Earth Observing System (AMSR-E), the Special Sensor Microwave/Imager (SSM/I), and the Tropical Rainfall Measuring Mission (TRMM) Microwave Imager (TMI) using the same OE configurations. This was later expanded to the Global Precipitation Measurement (GPM) Microwave Imager (GMI) (Duncan and Kummerow, 2016). The Colorado State University 1 D variational inversion algorithm (CSU 1DVAR) has been validated by comparing results with other independent products, showing that CSU 1DVAR can provide consistent results across a broad spectrum of sensors (Elsaesser and Kummerow, 2008; Duncan and Kummerow, 2016; Schulte and Kummerow, 2019; Schulte et al., 2020). A conceptually similar OE method is employed in the Microwave Integrated Retrieval System (MiRS; Boukabara et al., 2011, 2013, 2018) designed to provide various atmospheric and surface parameters (skin temperature, surface emissivity, and profiles of temperature, water vapor, non-precipitating clouds, and precipitations) under all sky conditions over ocean and land surfaces. Due to its flexible structure, MiRS is used operationally at NOAA and supports measurements from multiple MW instruments, including the TMI, GMI, MHS, Atmospheric Microwave Sounding Unit (AMSU), SSM/I, Special Sensor Microwave Imager/Sounder (SSMI/S), and Advanced Technology Microwave Sounder (ATMS).

Infrared (IR) sounders, and especially hyperspectral IR sounders, while limited to clear sky conditions, have distinct advantages for deriving temperature and moisture profiles due to their sharper weighting functions, particularly in the upper troposphere when no clouds are present. Using MW measurements from AMSU-A and MHS plus IR observations from the Infrared Atmospheric Sounding Interferometer (IASI) on board the MetOp platforms, Aires (2011) and Aires et al. (2011, 2012) significantly reduced the errors of retrieving temperature and water vapor profiles under clear sky conditions over the ocean by comparing with retrievals using individual MW or IR instruments alone. Under the European Space Agency Water Vapour Climate Change Initiative project (Siddans et al., 2015; Siddans, 2019), Trent et al. (2023) validated 9.5 years of atmospheric profiles retrieved from MetOp MW and IR observations and showed that global biases of temperature and water vapor are within 0.5 K and 10 %, respectively, making the retrieval products an important climate data record.

In addition to MW and IR measurements on the MetOp platforms, Milstein and Blackwell (2016) also showed the advantages of using MW and IR spectral bands from the Atmospheric Infrared Sounder (AIRS) and AMSU on the Aqua satellite as well as from the Cross-Track Infrared Sounder (CrIS) and ATMS on the Suomi National Polar-orbiting Partnership satellite (Suomi NPP) for temperature and water vapor retrievals. The NOAA Unique CrIS/ATMS Processing System (NUCAPS; Gambacorta et al., 2012) was built specifically to retrieve global atmospheric profiles using MW sensors (AMSU, ATMS,

and MHS) and hyperspectral IR instruments (AIRS, CrIS, or IASI) under non-precipitating conditions with
up to 80 % effective cloud fraction. Sun et al. (2017) used radiosonde data to assess the sounding
products from NUCAPS, indicating small biases in the lower atmosphere for temperature profiles of
less than 0.5 K and less than 20 % for water vapor profiles. These profiles have been further improved
by Ma et al. (2021), who applied a neural network technique to enhance the retrieved atmospheric
profiles in NUCAPS products by using IR channels on the next-generation series of Geostationary
Operational Environmental Satellites (GOES-R; Schmit et al., 2008). The root-mean-square error of
retrieved temperature and humidity profiles in that study decreased by more than 30 % from the
surface up to 700 hPa. Thus, while it seems clear from these previous studies that merging IR and MW
soundings from the same platforms is beneficial, CubeSat sounders such as TEMPEST or the Time-
Resolved Observations of Precipitation structure and storm Intensity with a Constellation of Smallsats
(TROPICS; Blackwell et al., 2018) do not generally fly in tandem with hyperspectral IR sounders. In this
case, it is useful to examine if there are benefits to merging the stand-alone passive MW sensors with
geostationary IR sounding channels.
The Advanced Baseline Imager (ABI), on board the GOES-R satellite series, observes the full disk of the
Earth every 10 minutes (15 minutes prior to April 2019), measuring in the visible (VIS), near-IR, and IR
spectral bands with spatial resolutions from 0.5 to 2 km. Except for the ozone absorption band at 9.6
$\mu$m (ABI channel 12), ABI channels 8 to 16 (6.2 to 13.3 $\mu$m) have different degree of humidity
sensitivities and are suitable for deriving water vapor profiles with similar vertical resolution to the
operational MW sensors (Schmit et al., 2008; Goodman et al., 2019; Li et al., 2019). Due to the high
spatial and temporal resolutions from GOES-R ABI observations over large regions, the ABI sensor can
always be matched with stand-alone MW radiometers over the sensed hemisphere, as illustrated by
Ma et al. (2021). This study thus focuses on the enhancement in water vapor retrievals that may be
achieved when ABI IR sounding channels are added to the TEMPEST-D MW channels.
**2.    Data**
The TEMPEST-D satellite (Reising et al., 2018) was deployed from the International Space Station on
July 13, 2018, into the Low Earth Orbit. The initial orbit height was 400 km with a 51.6° inclination,
observing an 825 km wide swath from the initial height. The mission successfully demonstrated both
the maneuverability of CubeSats to fly in closely maintained formations as well as the calibration
stability of the MW radiometer (Berg et al., 2021). The TEMPEST-D passive MW radiometer scanned
Earth in a cross-track mode and measured five channels at 87, 164, 174, 178, and 181 GHz with quasi-
horizontal polarization, except for 87 GHz, which measured quasi-vertical polarization. The spatial
resolutions of TEMPEST-D at the nadir were 14 km at 164 to 181 GHz and 28 km at 87 GHz. While the
data is not complete due to difficulties with the data receiving station at Wallops Island, Virginia, USA,
all available TEMPEST-D datasets can be requested through the website https://tempest.colostate.edu.
TEMPEST-D was deorbited on June 22, 2021. A second copy of TEMPEST was launched on Dec. 21,
2021, and is operating on the International Space Station in conjunction with COWVR. Data is available
from the National Aeronautics and Space Administration (NASA) Physical Oceanography Distributed
Active Archive Center (PODAAC) housed at NASA's Jet Propulsion Laboratory. Because the instruments
and orbits are identical, the results presented here apply to both sensors.

The GOES-16 (Schmit et al., 2008; Li et al., 2019) is the first of the GOES-R series satellites and was launched on November 19, 2016, carrying several instruments, including ABI. GOES-16 replaces GOES-13 and is located at longitude 75.2°W in a geostationary orbit (35786 km altitude), observing from latitude 81.32°N to 81.32°S and from longitude 156.30°W to 6.30°E. This covers North and South America, the eastern Pacific Ocean, and the Atlantic Ocean to the west coast of Africa. The ABI sensor measures 16 spectral channels from VIS to IR bands (0.47 to 13.3 µm) with spatial resolutions ranging from 0.5 km at 0.64 µm to 2.0 km in the IR. The eight ABI water-vapor-sensitive channels at 6.2, 6.9, 7.3, 8.4, 10.3, 11.2, 12.3 and 13.3 µm are used to enhance the TEMPEST-D retrieved water vapor profiles. While the ABI window and $CO_2$ channels (8.4, 10.3, 11.2, 12.3 and 13.3 µm) have information that is similar with the TEMPEST window channels, more measurements provide more information content to help constrain retrievals in a way used in the hyperspectral IR retrievals (Aires 2011; Aires et al., 2011, 2012; Gambacorta et al., 2012; Siddans et al., 2015). To ensure spatial and temporal consistency between TEMPEST-D and the GOES-16, the nearest geolocated ABI full disk pixels from ABI Radiances (RadF), Clear Sky Masks (ACMF), Cloud Top Phase (ACTPF), and Cloud Top Pressure (CTPF) products are averaged to match the geolocated TEMPEST-D pixels in space and time. The GOES-16 products can be downloaded through the Comprehensive Large Array Data Stewardship System (CLASS). Although GOES-17 also covers parts of the TEMPEST-D operational period, its products are not used to avoid all issues related to the cooling system, as described in https://www.goes-r.gov/users/GOES-17-ABI-Performance.html.

Except for satellite observations and products mentioned above, auxiliary data, including surface wind speed and direction, surface pressure, surface skin temperature, and temperature profiles, are also used to constrain the retrievals. These are taken from the ERA5 (Hersbach et al., 2020), accessed through the website https://www.ecmwf.int/en/forecasts/dataset/ecmwf-reanalysis-v5. The hourly ERA5 data used in the study are 0.5° x 0.5° with 27 pressure levels from 1000 to 100 hPa. The vertical resolution (in pressure coordinates) consists of 25 hPa intervals from 1000 to 750 hPa, 50 hPa intervals from 750 to 250 hPa, and 25 hPa intervals from 250 to 100 hPa. One hour temporal resolution and 0.5° spatial resolution from ERA5 is used to define unobserved surface conditions as well as the temperature profiles. The auxiliary surface parameters and temperature profiles are linearly interpolated in space and time to match the TEMPEST-D observations. The interpolated ERA5 auxiliary data may not reflect the actual conditions at the satellite overpass location and time, so when compared with in situ measurements, retrievals may be degraded by using the non-representative auxiliary data.

## 3. Methods

In satellite remote sensing, OE is a widely utilized technique to retrieve atmospheric components (Rodgers, 2000; Elsaesser and Kummerow, 2008; Boukabara et al., 2011; Siddans et al., 2015; Duncan and Kummerow, 2016; Schulte and Kummerow, 2019; Schulte et al., 2020). In OE, the state parameters and measurement errors are all assumed to follow a Gaussian distribution, and the atmospheric states being retrieved, $x$, are optimally estimated by minimizing the cost function $J$,

$$J = (x - x_a)^T S_a^{-1} (x - x_a) + [y - f(x)]^T S_y^{-1} [y - f(x)], \qquad (1)$$


where $x_a$ is the a priori information about the state vector $x$, $y$ is the measurement vector, $f(x)$ is a
forward model simulating measurements for a given state $x$, $S_a$ is the covariance matrix of a priori, and
$S_y$ is the covariance matrix of measurement errors (Rodgers, 2000). The minimization of $J$ is achieved
by iteratively solving for the state vector $x$ using the Gauss-Newton method. Following Eq. 5.29 in
Rodgers (2000), the convergence criteria are achieved when

$$d_i^2 = (x_i - x_{i+1})^T \hat{S}^{-1}(x_i - x_{i+1}) \ll n, \qquad\qquad (2)$$

where $d$ measures the change in the state vector between $i$th and $i$th + 1 iteration, and $n$ is the
number of retrieved variables (levels of water vapor and/or layers of clouds in this study). The solution
is said to have converged when the residual is one tenth the number of the retrieved variables in the
study. This is consistent with the definition from Eq. (2) that the error weighted increment is much less
than the number of the retrieved variables. The a priori state vector $x_a$ is used as the initial guess at
the beginning of the iteration. The a priori information $x_a$ and its uncertainty $S_a$ are derived from
monthly ERA5 humidity and cloud profiles over the ocean; $x_a$ describes the mean state of the profiles,
and $S_a$ accounts for the variation of the states. If sky conditions are known from GOES-16 cloud masks,
$x_a$ and $S_a$ obtained from clear or cloudy conditions will be used in the retrievals, or otherwise, a priori
values computed from all-sky conditions will be used.

The state vector $x$ comprises the water vapor mixing ratio at different pressure levels and/or clouds.
The number of selected water vapor levels depends on the number of channels and the assumptions of
clouds. The selected water vapor levels are evenly distributed in pressure levels at 1000, 900, 800, 600,
and 400 hPa for TEMPEST only, and 1000, 950, 875, 800, 700, 600, 450, and 350 hPa when both
TEMPEST and ABI channels are used. The remaining water vapor levels are linearly interpolated.
Following previous studies (Schulte and Kummerow, 2019; Schulte et al., 2020), clouds are inserted
into single layers containing liquid and/or ice clouds in the profiles. Since passive MW sensors do not
have information about cloud top height, if clouds are assumed to be present, the state vector will
contain one layer of liquid and one layer of ice clouds with liquid cloud top at 900 hPa and ice cloud top
at 300 hPa. If cloud information is available from GOES-16 products, liquid clouds and/or ice clouds can
also be inserted following GOES-16 cloud information as listed in Table 1. The table allows for
experiments where the GOES-16 is used simply to determine if there are clouds in the field of view
(FOV) or the actual cloud properties. If GOES-16 is only used to make the clear or cloudy
determination, then the cloud fraction is set to 0 or 1, respectively. TEMPEST-D, by itself, has no ability
to retrieve the cloud fraction. If details of the cloud field are used, the cloud fraction is set accordingly.



Table 1. The retrieval configurations under clear and cloudy conditions with and without GOES-16
cloud information. ABI means using eight ABI channels 8, 9, 10, 11, 13, 14, 15 and 16 (6.2, 6.9, 7.3, 8.4,
10.3, 11.2, 12.3 and 13.3 μm). ABI_3W means using three ABI water-vapor-sounding channels 8, 9 and
10 (6.2, 6.9 and 7.3 μm). CF, CH, and CP represent cloud fraction, cloud height, and cloud phase,
respectively.

| Sensors | Using GOES-16 cloud products | |
| --- | --- | --- |
| | Clear sky | Cloudy sky |
| TEMPEST+ABI (13 channels) or TEMPEST+ABI_3W (8 channels) or TEMPEST (5 channels) | 1. No, set CF to 1 2. Yes, set CF to 0 | 1. No, set CF to 1 2. Yes, set CF from GOES-16 3. Yes, set CF, CH, and CP from GOES-16 |

The measurement error covariance matrix $S_y$ is derived from two uncertainty sources: the radiometer
and the forward model (Elsaesser and Kummerow, 2008; Duncan and Kummerow, 2016; Schulte and
Kummerow, 2019; Schulte et al., 2020). The noise equivalent differential temperature (NEDT) values
are represented as the radiometer measurement errors for each sensor channel. For TEMPEST from 87
to 181 GHz, the NEDT values are 0.20, 0.35, 0.55, 0.55, and 0.75 K, respectively, which are evaluated
between 275 and 315 K (Berg et al., 2021; Padmanabhan et al., 2021). The NEDT values of ABI are 0.1 K
for all ABI IR channels, except for band 16, which is 0.3 K, and are evaluated at 300 K (Goodman et al.,
2019; GOES-R Series, 2022). The forward model uncertainties are approximated by comparing
simulated satellite observations using full ERA5 profiles to degraded simulated measurements using
the assumptions made in the OE retrievals, as described above. While the radiative transfer model is
assumed to contain no errors, errors are introduced when complex water vapor profiles are replaced
by simplified water vapor profiles at the previous prescribed retrieval levels, and complex cloud vertical
profiles are replaced by single liquid and ice cloud layers containing the equivalent cloud water path.
The measurement error covariance matrix $S_y$ is then derived from the NEDT values and the estimated
forward model errors. Figures 1(a), 1(b), and 1(c) show the $S_y$ estimated from all, cloudy and clear
skies, respectively, based on oceanic ERA5 profiles. Since ERA5 profiles most often contain some
degree of clouds, Figs. 1(a) and 1(b) have similar patterns, and channels having similar water vapor
sensitivity are more correlated with each other. On the other hand, due to much lower atmospheric
absorption in the clear skies, the surface-sensitive TEMPEST channels (87 and 164 GHz) have higher
correlations among themselves as in Fig. 1(c), although with smaller overall $S_y$ values than in Figs. 1(a)
and 1(b).

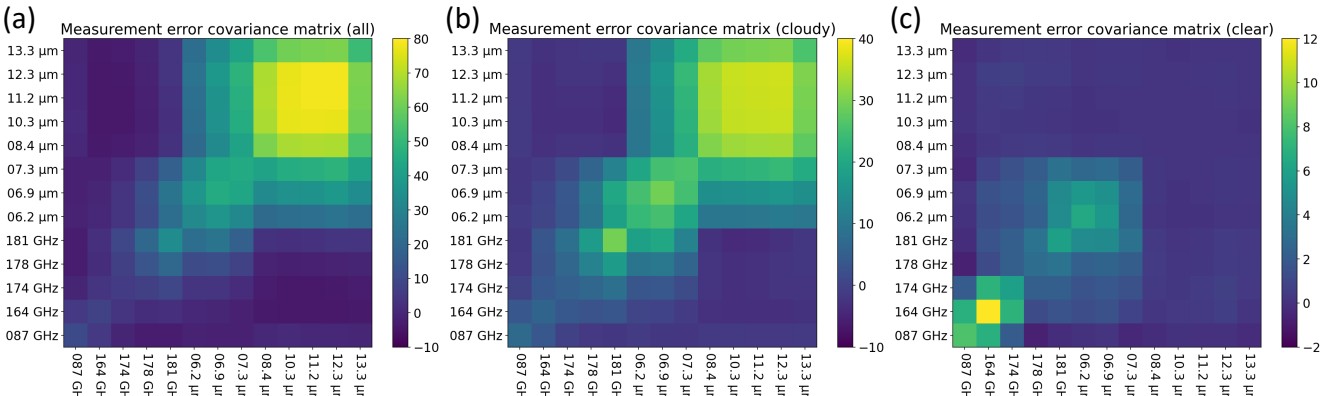


Figure 1. Measurement error covariance matrix $S_y$ for five TEMPEST-D MW and eight ABI IR channels derived from ERA5 profiles under (a) all sky, (b) cloudy sky, and (c) clear sky conditions over the ocean. The unit of the color is $K^2$.

The forward model is composed of two radiative transfer models: one simulates MW observations, and the other computes IR measurements. In the study, the Community Radiative Transfer Model (CRTM; Liu et al., 2012; Johnson et al., 2023) version 2.3.0 is used to calculate the observed brightness temperature for the ABI IR channels. The model can be downloaded through the website https://github.com/JCSDA/crtm. To simulate TEMPEST MW observations, an Eddington approximation, as described in Schulte and Kummerow (2019) and Schulte et al. (2020), is used. The Monochromatic Radiative Transfer Model (MonoRTM; https://github.com/AER-RC/monoRTM; Clough et al., 2005) is used to generate the atmospheric absorption while the ocean surface MW emissivity is computed using the FAST microwave Emissivity Model version 6 (FASTEM-6; Kazumori and English, 2015).

In the forward model, clouds are assumed to be homogeneously distributed in single layers. The cloud top pressure is 900 hPa for liquid clouds and 300 hPa for ice clouds if no cloud top heights are assigned from GOES-16 products, as described earlier. The CRTM default liquid and ice cloud optical properties are used to simulate IR brightness temperature with 12 and 30 μm effective radius for liquid and ice clouds, respectively. The MW optical properties of liquid clouds are generated by Lorenz-Mie theory (van de Hulst, 1957; Bohren and Huffman, 1998), assuming the droplet is spherical with a radius of 12 μm and is monodisperse in particle size distribution (PSD). The radiative properties of ice clouds in the MW spectrum are computed using the single-scattering property databases for non-spherical ice particles from Liu (2008) and Nowell et al. (2013) following the analysis of Schulte and Kummerow (2019). The databases are derived by the discrete-dipole approximation method (Draine and Flatau, 1994). The microphysical properties of ice clouds used to derive the scattering properties are assumed to have the PSD from Field et al. (2007) with a constant density of 100 g/cm$^3$ and have ice habits: 6 bullet rosettes (crystal size $<$ 800 μm) and aggregates of 400 μm rosettes (crystal size $\geq$ 800 μm). The spectral inconsistency of cloud optical properties and miss-representing ice clouds can be two of the major error sources in radiative transfer simulations (Kulie et al., 2010; Yang et al., 2018; Ringerud et al., 2019; Schulte and Kummerow, 2019; Yi et al., 2020), but are not considered here.

The monthly means and variability of water vapor mixing ratios from ERA5 above 200 hPa are extremely small, as shown in Fig. 2. The sensor responses to these small amounts of stratospheric water vapor are less than the noise of 0.2 to 0.75 K for TEMPEST and 0.1 to 0.3 K for ABI. Therefore, the water vapor mixing ratio was set to the monthly mean climatology above 200 hPa and is not retrieved explicitly with the available channels.

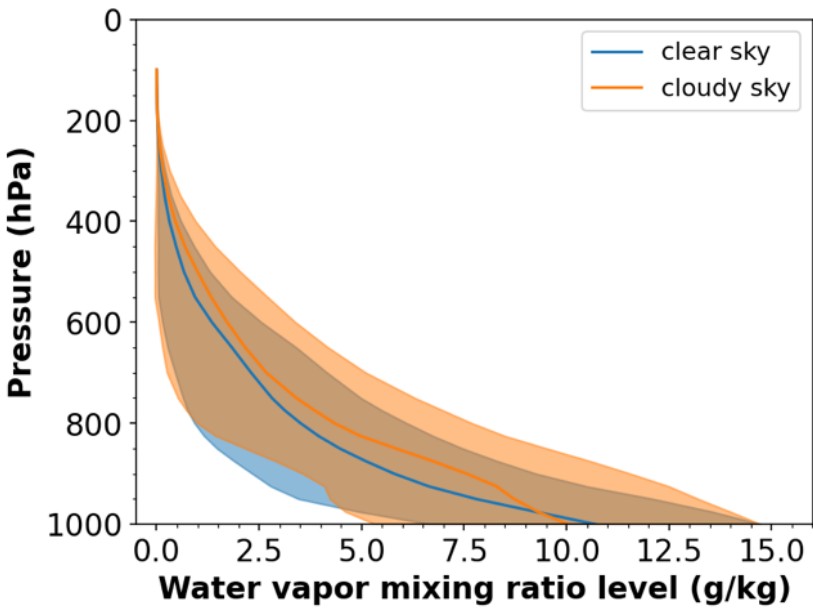

Figure 2. Monthly mean and standard deviation (σ) of water vapor profiles under clear and cloudy
conditions over the ocean between ± 60° latitudes from ERA5 in May 2020. Blue color represents
water vapor in clear skies, while orange color shows water vapor in cloudy skies. Solid lines are mean
water vapor profiles, and shaded areas are standard deviations.
With the model configuration described above and a priori atmospheric temperature and water vapor
profiles from ERA5 shown in Figs. 3(a) and 3(b), the sensitivity of water vapor to five TEMPEST-D MW
channels and eight ABI IR bands is represented by the clear sky Jacobians shown in Fig. 3(c), and in the
cloudy sky Fig. 3(d) presents the Jacobians of water vapor and clouds. For humidity, all TEMPEST MW
and ABI IR channels have different degrees of sensitivity along the altitude axis. In clear or cloudy skies,
three ABI water-vapor-sounding channels (6.2 to 7.3 μm) only provide signals for the upper
atmosphere. However, signals of water vapor are sensed from the surface to the top of the
atmosphere by the TEMPEST MW bands under both clear and cloudy conditions and by ABI window
and $CO_2$ bands (8.4 to 13.3 μm) in the clear sky. Although the water vapor sensitivity is substantially
reduced under liquid clouds in ABI window and $CO_2$ bands, TEMPEST 87 and 164 GHz window bands
have significant sensitivity to water vapor and liquid clouds through the entire lower atmosphere.
Except for the TEMPEST 87 GHz band, all remaining TEMPEST channels have sensitivity to ice clouds.
Overall, as also shown in the studies mentioned in the introduction (Aires, 2011; Milstein and
Blackwell, 2016; Sun et al., 2017; Ma et al., 2021; Trent et al., 2023), Figs. 3(c) and 3(d) demonstrate
the advantage of merging IR and MW spectral bands in soundings: MW channels have humidity signals
under cloudy conditions, IR water-vapor-sounding bands provide extra information about the upper
atmosphere, and IR window and $CO_2$ channels have humidity sensitivity in the clear sky.

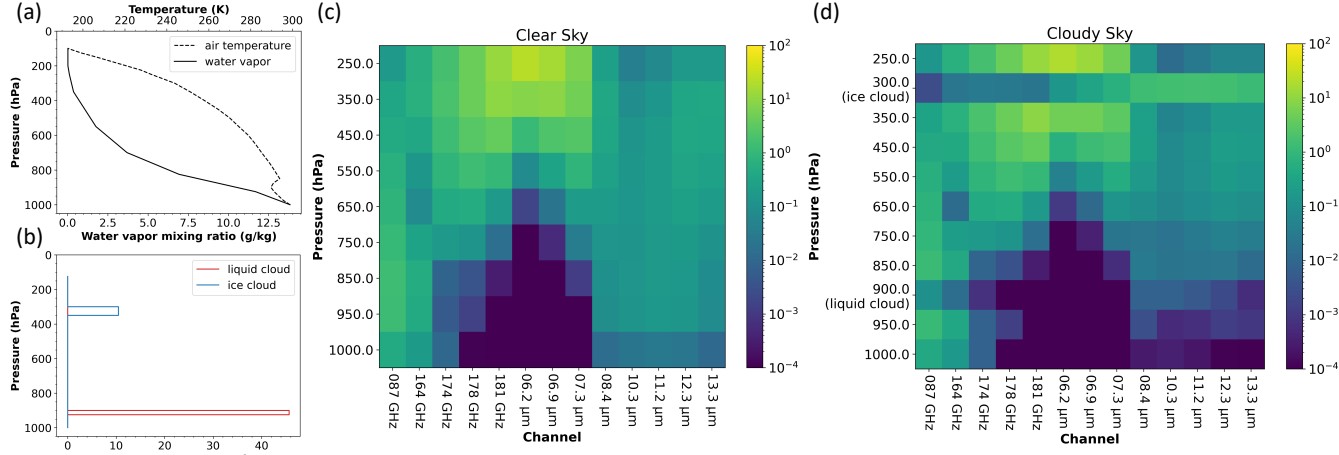

Figure 3. An example of water vapor and cloud Jacobians and the ERA5 profiles over the ocean used to
compute the Jacobians. (a) Profiles of air temperature and water vapor mixing ratio. (b) Liquid and ice
cloud layers. (c) Water vapor Jacobians from 250 to 1000 hPa in the clear sky as a function of sensor
channels (TEMPEST-D from 87 to 181 GHz and ABI from 6.2 to 13.3 μm). (d) The same as (c) but for
water vapor Jacobians from 250 to 1000 hPa and Jacobians of liquid (cloud top at 900 hPa) and ice
(cloud top at 300 hPa) clouds in the cloudy sky. The unit of the color for water vapor Jacobians is
K/g/kg, and for liquid and ice cloud Jacobians is K/g/m$^2$.

Given the frequent observation from GOES-R ABI, the data can be readily merged with TEMPEST-D.
Figure 4 shows the overlap of the two sensors over the ocean. Gaps between MW orbits, as well as
cloudy regions where ABI detects clouds, are evident in both images. Even though ABI cannot be used
for sounding in cloudy atmospheres, using the ABI cloud products can still provide retrievals some prior
knowledge about clouds (cloud fraction, phase, and height), which will be shown to positively impact
the TEMPEST-D MW retrievals. The next section will explore retrieval sensitivities under clear and
cloudy conditions using synthetic TEMPEST-D and ABI observations simulated from ERA5 profiles.
Retrieved water vapor profiles are then validated against in situ radiosonde humidity measurements
under different retrieval assumptions, as listed in Table 1.

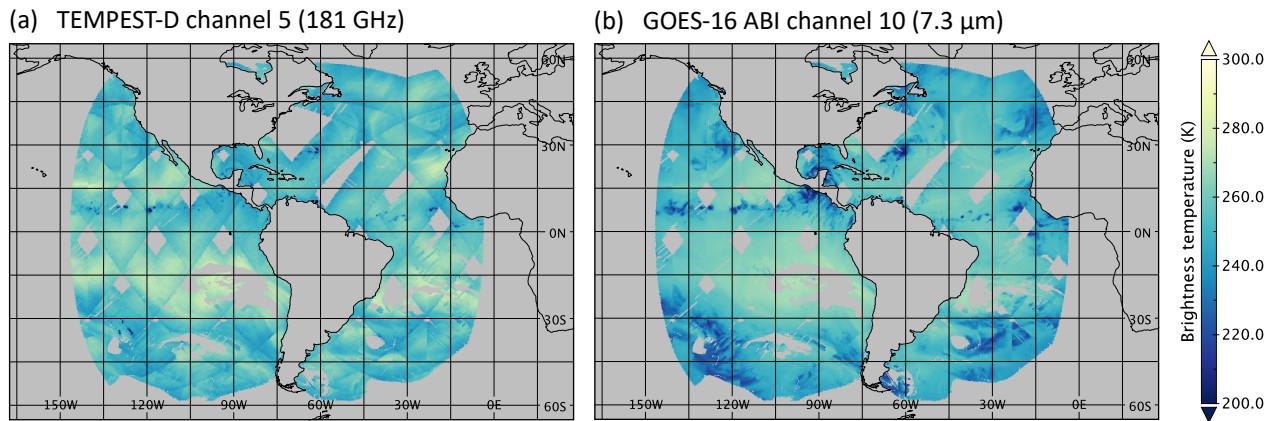


Figure 4. Collocated TEMPEST-D and GOES-16 ABI observations over the ocean on 2020/06/01 for (a) TEMPEST-D channel 5 (181 GHz) and for (b) ABI channel 10 (7.3 μm).

## 4.    Results

### 4.1. Sensitivity Tests

Observations for the TEMPEST five (87, 164, 174, 178, and 181 GHz) and ABI eight (6.2, 6.9, 7.3, 8.4. 10.3, 11.2, 12.3, and 13.3 μm) channels are simulated using temperature, humidity, cloud profiles, surface temperature, and surface wind speed and direction from ERA5 over the ocean with viewing angles corresponding to TEMPEST and ABI instruments respectively. All data corresponds to May 27, 2020. Since the true states from the ERA5 data are known, the retrieval accuracy can be evaluated using the computed observed brightness temperature under different scenarios.

#### 4.1.1.    Case studies

Two cases are used to illustrate the humidity retrievals, first using only the TEMPEST sensor, then adding three ABI water-vapor-sounding channels, and then using eight ABI bands in clear and cloudy sky scenes. These are shown in Fig. 5. While the retrieved profiles do not change dramatically, the additional ABI channels can be seen to improve the mid-tropospheric biases, as shown in Figs. 5(b) and 5(d), especially using eight ABI bands in Fig. 5(b) and adding three ABI water vapor channels in Fig. 5(d). Although the retrieved water vapor profiles are over- and under-estimated along the height when compared to the true ERA5 values, Figs. 5(a) and 5(b) reveal that the retrievals using eight extra ABI IR channels improve significantly with respect to both bias and standard deviation under clear condition where five ABI window and $CO_2$ bands provide additional signal from the lower atmosphere in addition to three ABI water vapor channels giving upper atmosphere information shown in Fig. 3(c). In the cloudy scene, since ABI window and $CO_2$ channels are heavily affected by clouds as Fig. 3(d), Figs. 5(c) and 5(d) show that water vapor retrievals are slightly degraded by using eight ABI channels than by adding three ABI water vapor bands, which improve retrievals above the 800 hPa level where the ABI water-vapor-sounding channels are expected to add the most information. While overall biases and standard deviations also decrease for both examples, it is apparent that ABI has little influence over the low-level water vapor and that most of the improvement actually comes from the mid to upper troposphere.

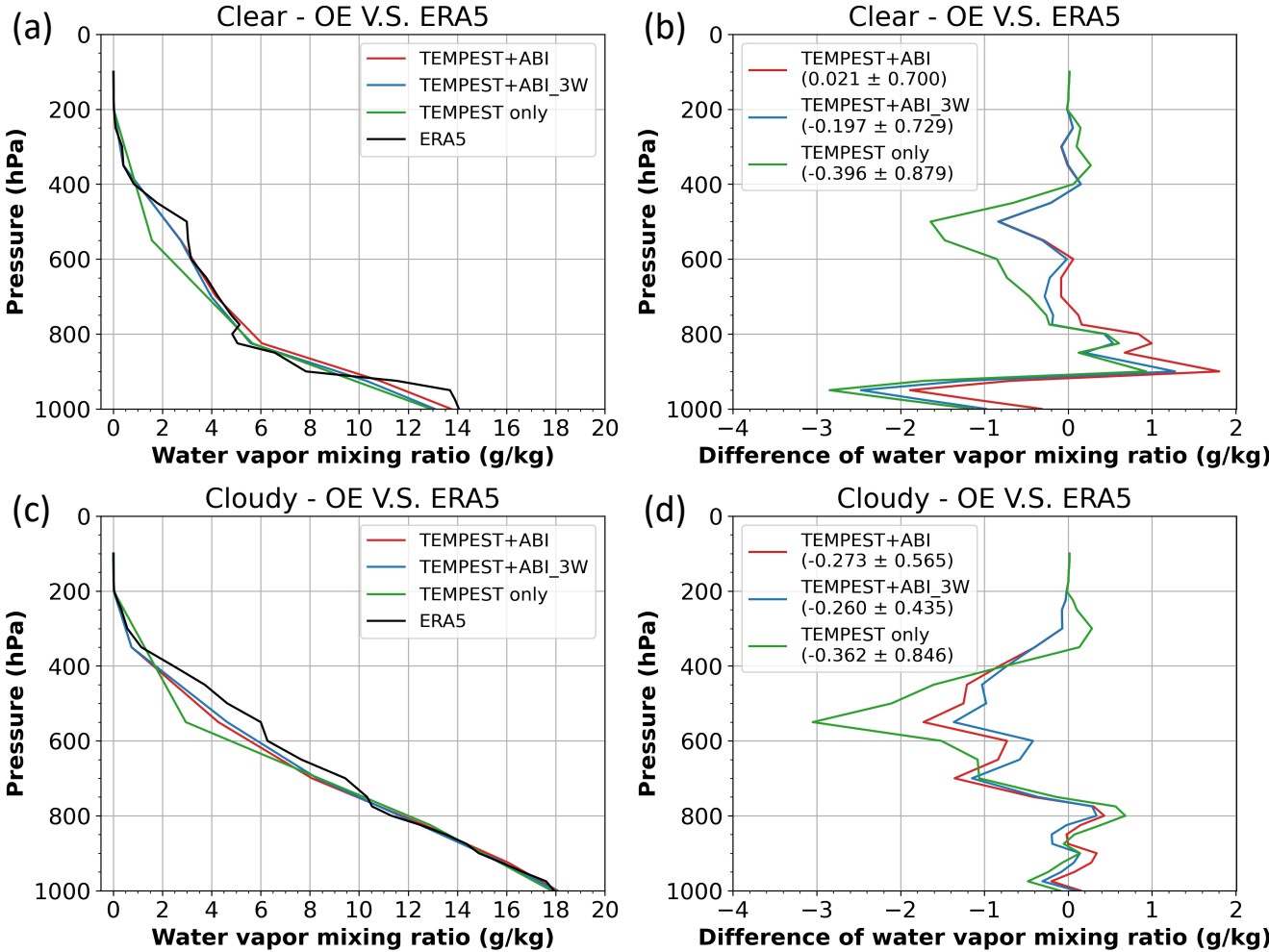

Figure 5. Two selected cases of retrieved water vapor profiles using the synthetic observations from
ERA5 over the ocean on 2020/05/27 and using all-sky a priori. Figures (a) and (b) show retrievals under
clear conditions, while cloudy retrievals are presented in Figures (c) and (d). Figures (a) and (c) show
the retrieved and ERA5 humidity profiles and the corresponding comparisons between retrievals and
ERA5 (retrievals minus ERA5) are presented in Figures (b) and (d). The solid black lines are water vapor
profiles from ERA5. The solid red lines are water vapor retrievals using five TEMPEST and eight ABI
combined channels, and retrievals using TEMPEST and three ABI water vapor bands are the solid blue
lines. The solid green lines are retrievals using the TEMPEST sensor. The number in the parentheses is
the bias ± standard deviation of the whole profile.
**4.1.2. Statistics**
Comparisons of humidity retrievals using merged five TEMPEST MW bands and three or eight ABI-
sounding channels (6.2 to 13.3 μm) versus using only the TEMPEST sensor are performed for 1000
randomly selected clear or cloudy sky cases. Based on the GOES-16 ABI cloud mask, there are about
1200 clear sky and 8400 cloudy pixels successfully collocated with TEMPEST on May 27, 2020.
Randomly selecting 1000 samples from both clear and cloudy pixels allows fair statistical comparisons
between clear and cloudy regions. The statistics are found independent of how the 1000 samples are
randomly selected. Results in clear skies are shown in Fig. 6. As with the case studies, adding ABI
channels clearly reduces layer biases and random errors in the retrieved water vapor profiles. Errors in
the retrieved water vapor above 800 hPa are significantly smaller when using the five MW bands from
TEMPEST in combination with the ABI channels. Particularly, among these three retrieval
configurations, with the additional information provided by five ABI window and $CO_2$ channels (8.4 to
13.3 $\mu$m), using eight ABI bands in the water vapor retrievals has the least overall biases and standard
deviations and improves retrievals around the surface, where the biases are less than 1 g/kg for using
eight ABI and five TEMPEST bands in Fig. 6(a) and are about 1.2 to 1.4 g/kg for using five TEMPEST
with/without three ABI water vapor channels in Fig. 6(b) and 6(c). While the overall water vapor biases
and standard deviations under clear conditions are reduced only slightly from (-0.149 $\pm$ 1.127 g/kg) for
TEMPEST only to (-0.128 $\pm$ 1.022 g/kg) for TEMPEST+ABI_3W, much larger reductions can be seen in
the TEMPEST+ABI retrievals (-0.014 $\pm$ 0.944 g/kg) and in the layer values shown in Fig. 6 – starting at
900 hPa and extending all the way to 300 hPa.


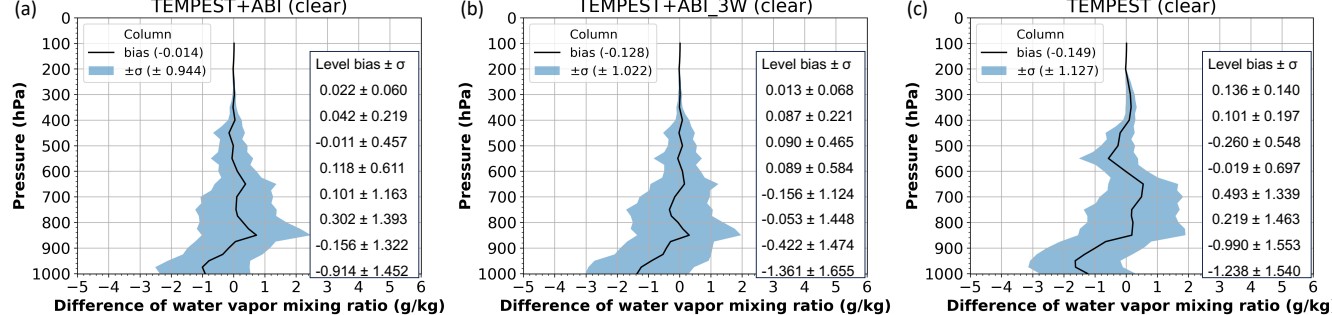

Figure 6. Sensitivity tests of retrieving water vapor profiles using the synthetic measurement from
ERA5 under clear conditions over the ocean on 2020/05/27 and using all-sky a priori. Figure (a) shows
retrievals using thirteen TEMPEST and ABI combined channels, Figure (b) presents retrievals using five
TEMPEST and three ABI water vapor channels, and retrievals using only TEMPEST channels are for
Figure (c). Figures (a) to (c) show the difference in water vapor mixing ratio from 1000 randomly
selected profiles between retrievals and ERA5 (retrievals minus ERA5) along the height. The solid black
lines are the bias value, and the blue shade area is the standard deviation ($\sigma$). The included table
quantifies the retrieval performance from 300 to 1000 hPa for every 100 hPa.


Similarly, the accuracy of humidity retrievals from 1000 randomly selected cloudy cases using three
different sensor configurations is shown in Figs. 7(a) to 7(c). Consistent with the case study and clear
sky cases shown in Fig. 6, adding ABI IR channels to the retrievals also reduces biases in the mid-
tropospheric layers for cloudy scenes. Due to the lack of sensitivity of ABI channels to the lower
atmosphere, as shown in Fig. 3(d), the performance of water vapor retrievals around the surface shows
only a negligible improvement in cloudy skies. While the column metrics show unbiased results with or
without ABI, the standard deviation of retrieval errors is larger when using TEMPEST-only retrievals
(1.022 g/kg) than using merged TEMPEST and three or eight ABI channels (0.949 or 0.898 g/kg).
Quantitative comparisons of the vertical profiles in Figs. 7(a) to 7(c) again reveal that the layer biases
are significantly reduced in the TEMPEST+ABI and TEMPEST+ABI_3W retrievals relative to TEMPEST
alone, reducing the individual layer biases by approximately 50 % (although not uniformly in all layers).
The overall biases are smaller than in the clear case. The latter is explained by the fact that the all-sky a
priori guess comes from the climatology of ERA5 profiles for the month, and these profiles
overwhelmingly contain clouds. The cloudy retrieval is thus less biased in the initial iteration, while the
clear retrievals must adjust the first guess to correspond to drier conditions when the atmosphere is
cloud-free. Standard deviations are slightly larger for cloudy scenes, as should be expected from a
more complex retrieval.

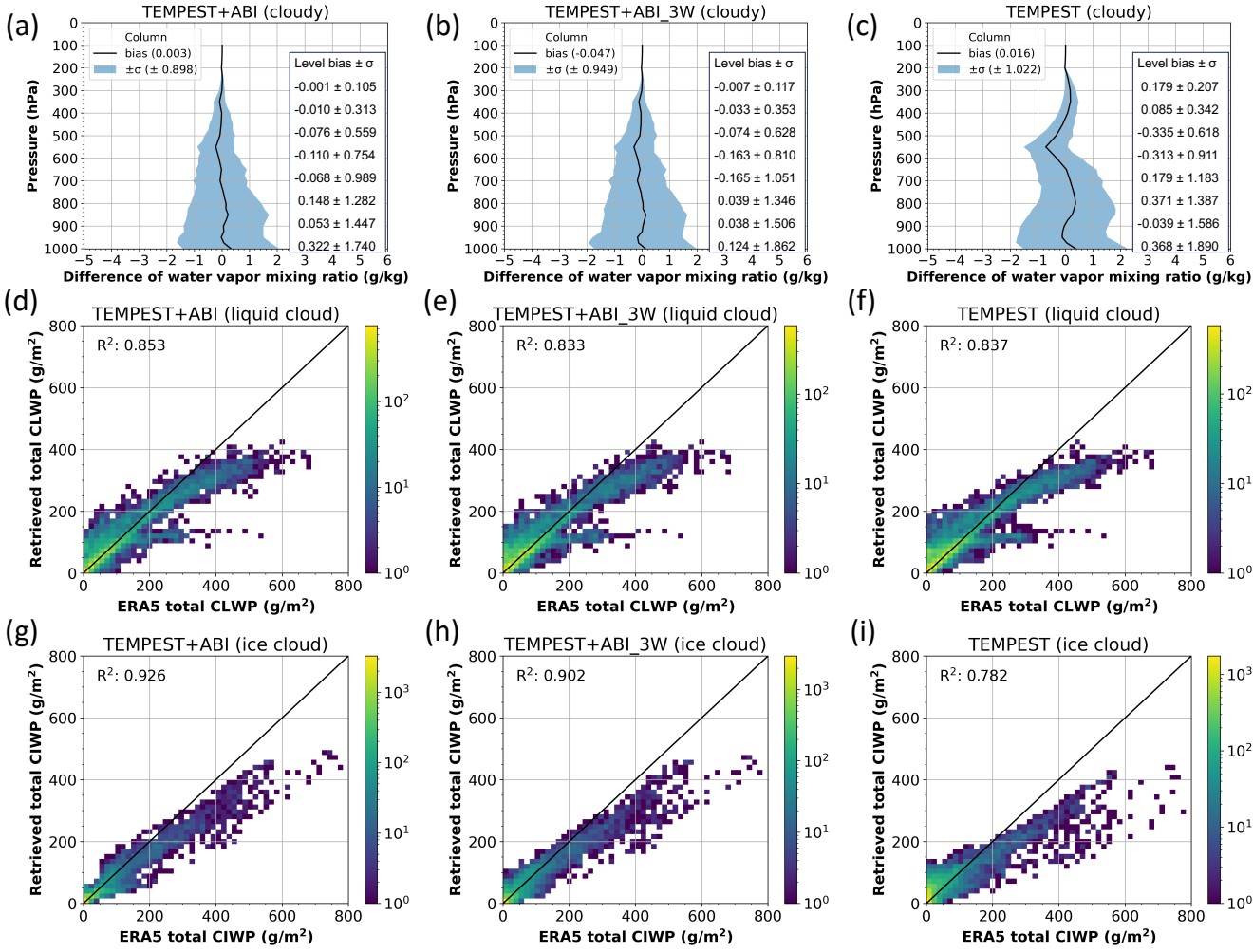

Figure 7. Sensitivity tests of retrievals of water vapor, liquid and ice clouds using synthetic observations
from ERA5 under cloudy conditions over the ocean on 2020/05/27 and using all-sky a priori. Figures (a),
(d), and (g) show retrievals using TEMPEST and eight ABI combined channels, Figures (b), (e), and (h)
present retrievals using merged TEMPEST and three ABI water vapor channels, and retrievals using
only TEMPEST channels are for Figures (c), (f), and (i). Figures (a) to (c) show the difference in water
vapor mixing ratio from 1000 randomly selected profiles between retrievals and ERA5 (retrievals minus
ERA5) along the height. The solid black lines are the bias value, and the blue shade area is the standard
deviation (σ). The included table quantifies the retrieval performance from 300 to 1000 hPa for every
100 hPa. Figures (d) to (f) are two-dimensional histograms of retrieved and ERA5 total cloud liquid
water path from 8000 randomly selected cases (total number of cloudy pixels is about 8400). $R^2$ is the
coefficient of determination. Color means the number of samples; the solid black lines are the one-to-
one lines. Figures (g) to (i) are the same as Figures (d) to (f) but for the total cloud ice water path.
The performance of liquid and ice cloud retrievals is shown in Figs. 7(d) to 7(i). Compared with the
cloud liquid water path from ERA5, the liquid cloud retrievals do not improve after incorporating three
more ABI water-vapor-sounding channels, shown in Figs. 7(e) and 7(f), as the cloud liquid water path
signal is confined almost entirely to the 87 and 164 GHz channels of TEMPEST-D. The sensitivity to
liquid clouds with and without three ABI channels is similar, with $R^2$ values about 0.83. However, given
additional cloud sensitive channels from five ABI window and $CO_2$ bands, liquid cloud retrievals are
slightly improved by using TEMPEST+ABI, as the $R^2$ values increase from about 0.83 to 0.85. Since ice
clouds are at a higher altitude and interact with window and $CO_2$ channels as well as the water-vapor-
sounding channels, the 164 to 181 GHz TEMPEST and 6.2 to 13.3 μm ABI channels have different
degrees of sensitivity, as shown in Fig. 3(d). Adding ABI channels has larger impacts on the retrieved ice
clouds, as the $R^2$ values increase from 0.782 using only TEMPEST bands to over 0.9 using eight or three
combined channels from TEMPEST and ABI. Especially, due to strong sensitivity from ABI channels 8.4
to 13.3 μm, merging five TEMPEST and eight ABI channels gives the best ice cloud retrievals ($R^2$ value is
about 0.93) among three retrieval configurations and significantly constrains retrieved ice water path
with less than 50 g/m$^2$. Overall, the retrieved liquid and ice clouds are all underestimated compared
with the ERA5 profiles. For liquid clouds, this is simply due to the saturation of the cloud water
emission signal at roughly 300 to 400 g/m$^2$ with the available channels. For ice clouds, the primary
signal is a brightness temperature depression due to scattering. While this signal does not saturate,
thicker ice clouds (> 300 to 400 g/m$^2$) are often found in conjunction with liquid clouds in ERA5, leading
to brightness temperature signatures that are more difficult to untangle.
**4.2. Independent Validation**
While the preceding section focused on synthetic brightness temperatures generated from ERA5
profiles, this section uses radiosonde data to validate retrievals from actual observations. The
Integrated Global Radiosonde Archive (IGRA) has collected and quality-controlled in situ observations
from over 2,800 global stations since 1905, providing vertical profiles of pressure, temperature,
humidity, and wind speed and direction. The IGRA dataset can be accessed at
https://www.ncei.noaa.gov/products/weather-balloon/integrated-global-radiosonde-archive. The
IGRA dataset used in the study is version 2.2 and is collocated with TEMPEST-D and GOES-16 ABI
observations from 2019 to 2020. To ensure consistency in collocated cases, the observations from
these three datasets are all within 1 hour and 1 degree latitude/longitude. Because the OE retrieval
discussed here is limited to oceans, the radiosondes used in this study are limited to coastal regions. To
avoid surface contaminations, the collocated TEMPEST-D measurements are moved over the ocean to
ensure that ~30 km (the sensor FOV) in all directions of the TEMPEST-D pixel is free of land. The
displaced footprints must have the same cloud conditions (clear sky or cloudy) as determined by GOES-
16 cloud products at the radiosonde location to ensure these locations are under similar atmospheric
conditions. There are 19 collocated coastal IGRA stations in the GOES-16 FOV, as shown in Fig. 8. The
collocated IGRA sites are around North America and the Caribbean Sea. Given GOES-16 cloud
information, there are 104 collocated cases, of which 10 cases are cloud-free, and 94 cases are under
different degrees of cloudy skies, as shown in Fig. 9. The limited number of coincident samples is due
to infrequent TEMPEST-D overpasses coupled with infrequent (twice daily) radiosonde launches and
frequent data downlink problems of TEMPEST-D, leaving only this limited set of radiosondes to
compare to.

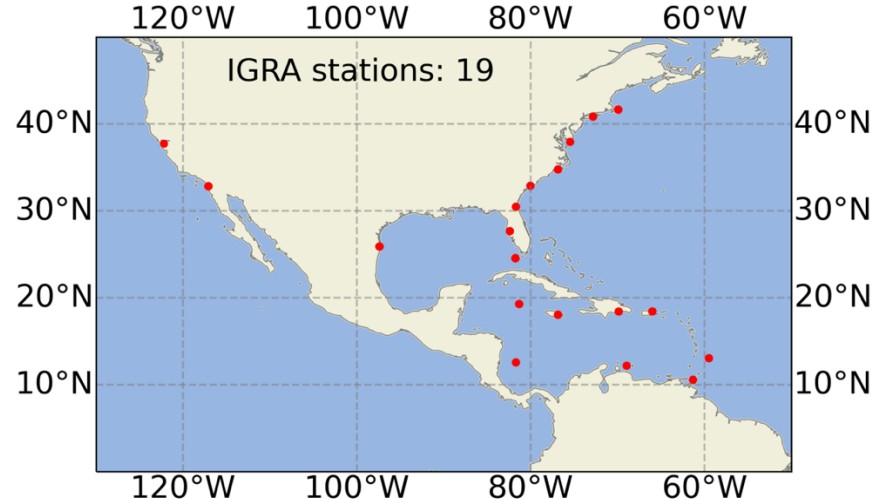

Figure 8. Map of collocated IGRA stations. The total number of collocated sites is 19, as marked in the
red circle dots.

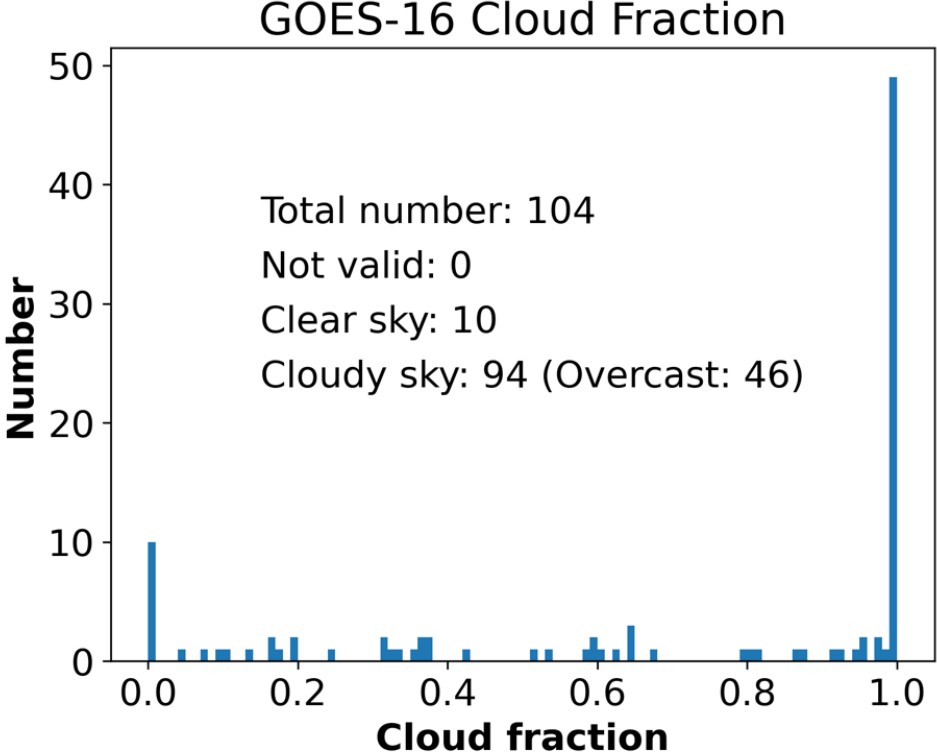

Figure 9. The histogram of GOES-16 derived cloud fraction at the collocated locations. The total number of collocated cases is 104, including 10 clear and 94 cloudy cases.

With additional cloud information from GOES-16 products, water vapor retrievals are validated with various levels of cloud information from the geostationary observations, as described in Table 1. The most significant difference is that the algorithm does not retrieve clouds when the area is cloud-free (as determined by ABI's cloud mask) and uses observations from all channels to retrieve water vapor profiles only. Figure 10 shows the error in the retrieved water vapor profiles in clear skies, with biases and standard deviations of column errors listed in Table 2. Only nine cases converged among ten clear sky cases under four different retrieval settings for using only TEMPEST bands and merged TEMPEST and ABI three water vapor channels; using five TEMPEST and eight ABI bands has slightly reduced the retrieval rate, which is eight out of ten cases. Experiments are performed with and without GOES-16 information. If GOES-16 cloud products are not used, the cloud fraction is set to 1.0, implying that clouds covering the FOV are possible, although the retrieval can set the cloud water path to zero. The convergence criteria from Eq. (2) are set to 0.8 for retrievals using TEMPEST-D and ABI three or eight channels and are 0.5 for using TEMPEST-D five bands, as mentioned in section 3 (either 5 or 8 layers of clouds/water vapor in this case).

Table 2. Compared with IGRA radiosonde observations, the column bias and standard deviation of retrieved water vapor mixing ratio under the clear sky conditions. The statistic values are evaluated based on all converged eight clear sky cases for the TEMPEST+ABI sensor configuration and nine clear sky cases for using TEMPEST and TEMPEST+ABI_3W channels. CF means cloud fraction.


| Sensors | Using GOES-16 cloud products | |
| --- | --- | --- |
| | No<br>set CF to 1 | Yes<br>set CF to 0 |
| TEMPEST+ABI<br>(13 channels) | -0.070 ± 1.085 g/kg | 0.476 ± 1.055 g/kg |
| TEMPEST+ABI_3W<br>(8 channels) | -0.201 ± 1.029 g/kg | 0.501 ± 1.071 g/kg |
| TEMPEST<br>(5 channels) | -0.124 ± 1.156 g/kg | 0.510 ± 1.078 g/kg |


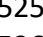

Figure 10. The water vapor mixing ratio difference between retrievals and radiosonde measurement
(retrievals minus IGRA) in the GOES-16 observed clear skies. Retrievals use thirteen bands from
TEMPEST-D and GOES-16 ABI in Figures (a) and (d), use five TEMPEST-D and three ABI water-vapor-
sounding channels in Figures (b) and (e), and use only TEMPEST-D channels in Figures (c) and (f).
Retrievals in Figures (a) to (c) assume existing liquid and ice clouds with cloud fraction = 1 and use all-
sky a priori, and retrievals in Figures (d) to (f) set no clouds with cloud fraction = 0 and use clear sky a
priori. In the retrievals, the biases of the water vapor a priori information derived from all-sky
conditions are shown in Figure (g), and obtained from clear skies are presented in Figure (h). The solid
black lines are the bias value, and the blue shade regions indicate the standard deviation ($\sigma$). The
included table quantifies the retrieval performance from 300 to 1000 hPa for every 100 hPa. The
number in the parentheses indicates the number of all converged cases out of all clear sky cases. G16
means GOES-16 products, and L+I indicates liquid and ice clouds.
The additional eight (6.2 to 13.3 µm) as well as three (6.2 to 7.3 µm) channels from ABI help to
constrain water vapor profiles, as shown in the reduced column error standard deviations as well as
the layer biases and standard deviations, although the differences are smaller than they were with the
simulated results. Compared with TEMPEST-only (Figs. 10(c) and 10(f)), the retrieved water vapor
profiles above 800 hPa are visibly less biased after including eight (Figs. 10(a) and 10(d)) or three (Figs.
10(b) and 10(e)) ABI channels. The overall statistics are not as impressive because much of the water
vapor is in the 1000 to 800 hPa layer, which is not improved by additional three ABI water-vapor-
sounding channels. However, with extra information from five ABI window and $CO_2$ bands, water vapor
retrievals have slightly improvement around the surface, leading to smaller entire retrieval biases and
standard deviations among these three sensor configurations. Figures 11(a) to 11(c) present the
erroneous retrieved liquid and ice clouds under the clear conditions corresponding to Figs. 10(a) to
10(c), respectively. No clouds are estimated in retrievals in Figs. 10(d) to 10(f), as this information is
taken from the IR channels. Because parts of the water vapor signals are falsely attributed to clouds,
retrieved water vapor profiles are underestimated when clouds are derived, as in Figs. 10(a) to 10(c)
and 11. On the other hand, retrieved water vapor profiles are overestimated in Figs. 10(d) to 10(f)
when the scene is forced to be cloud-free based on ABI information. We speculate that, as with the
synthetic retrievals, the bias from ERA5 information in Fig. 10(h) under clear sky assumptions is even
larger than if all sky ERA5 a priori in Fig. 10(g) is used. This leads to even larger biases in the initial
iteration, which the retrievals can only partially correct without adding small amounts of cloud water
to the scene. Conversely, it is also possible that the small number of cases (8 or 9) simply are not
representative.

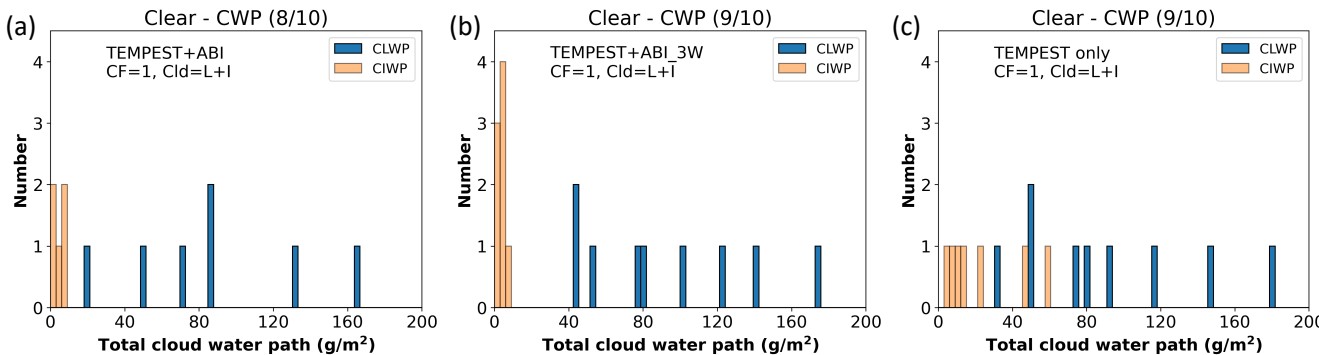

Figure 11. Retrieved total cloud water path for liquid and ice clouds in the clear sky cases with no cloud
information from GOES-16. Retrievals in Figure (a) and (b), in addition to five TEMPEST channels, use
eight ABI channels and three ABI water vapor bands, respectively, and use only TEMPEST-D channels
for Figure (c). The number in the parentheses indicates the number of all converged cases among all
clear sky cases. L+I indicates liquid and ice clouds.


Water vapor retrieval errors under cloudy conditions for various assumptions of cloud knowledge are
presented in Fig. 12, with the corresponding bias and standard deviation of column errors listed in
Table 3. Although cases used in Table 3 and Fig. 12 have all ABI and TEMPEST-D observations and all
cloud information, this is not the case for all other pixels. Therefore, Table 3 and Figure 12 show the
possible results from nine different retrieval configurations using different degrees of cloud status and
using TEMPEST-only or with measurements from eight or three ABI channels. The retrieval
configurations in cloudy cases are listed in Table 1. Due to lack of humidity sensitivity of ABI window
and $CO_2$ bands below clouds as Fig. 3(d), in comparisons with adding three ABI water-vapor-sounding
channels, using eight ABI bands doesn't improve water vapor retrievals and has much less retrieval
rate. Retrievals in Figs. 12(a) to 12(c) have no information about clouds. In contrast, Figs. 12(d) to 12(i)
show results with different degrees of knowledge about clouds from ABI. Figures 12(d) to 12(f) use only
cloud fractions. In the scenarios of no cloud information from ABI in Figs 12(a) to 12(c), water vapor
retrievals using TEMPEST+ABI and TEMPEST+ABI_3W have improvement above 500 hPa, between 700
and 800 hPa, and around the surface. When only cloud fraction is available from GOES-16 cloud
products, Figs 12(d) to 12(f) show that adding eight or three ABI bands improves overall water vapor
retrievals except for around 900 hPa. If the cloud fraction, cloud height, and cloud phase are all
available from the cloud products as in Figs 12(g) to 12(i), water vapor retrievals using different
degrees of ABI measurements have improvement around 300, 400, and 600 hPa and have minor or no
improvement on the other levels. In general, when retrievals use the same cloud status, column
average water vapor retrieval biases using TEMPEST and ABI observations are smaller than using
TEMPEST-only measurements, as in comparisons among Figs 12(a) to 12(c), Figs 12(d) to 12(f), and Figs
12(g) to 12(i). While column average water vapor retrievals do not improve significantly by adding
cloud fraction information, when cloud fractions are specified, quantitative comparisons show some
improvements between 500 and 700 hPa and around the surface for TEMPEST+ABI retrievals in Figs.
12(a) and 12(d) and for TEMPEST+ABI_3W retrievals in Figs. 12(b) and 12(e), and present some
improvements above 400 hPa and around 600 hPa and the surface for TEMPEST-only retrievals in Figs.
12(c) and 12(f).

Additional cloud information in the form of cloud fraction, cloud height, and cloud phase from GOES-16
products are shown in Figs. 12(g) to 12(i). When retrievals use more cloud information from GOES-16
(cloud fraction, height, and phase), water vapor retrieval biases shown in Fig. 12(h) are about half of
the biases in Figs. 12(b) and 12(e) around 600 hPa and shown in Fig. 12(i) are improved above 700 hPa
except for around 600 hPa compared with Figs. 12(c) and 12(f), but retrievals have no or minor
improvements above 700 hPa in Fig. 12(g) compared with Figs. 12(a) and 12(d). Water vapor retrievals
around lower layers in Figs. 12(g) to 12(i) show larger biases and little difference among using only
TEMPEST, TEMPEST+ABI_3W or TEMPEST+ABI. In cloudy conditions, the only channels with sensitivity
to the low-level water vapor are the TEMPEST 87 and 164 GHz channels, as shown in Fig. 3(d).
However, some overfitting appears to be taking place between 700 and 1000 hPa. The authors
speculate that the ice scattering properties assumed in the retrieval's forward model may cause excess
depression at 87 and 164GHz channels, which in turn, requires the algorithm to increase the cloud
water and water vapor to match the brightness temperatures in those channels. Meanwhile, since MW
and IR have different sensitivity to the clouds, the cloud properties obtained from ABI cloud products
are derived from VIS/IR bands (Goodman et al., 2019) may not be representative to more cloud
transparent MW channels, adding more uncertainties in retrievals.
The water vapor retrieval errors are further decomposed by cloud fraction from GOES-16, shown in Fig.
13, using various retrieval configurations shown in Table 1 under cloudy conditions. Since not enough
retrievals are obtained by TEMPEST+ABI configurations, Figure 13 only presents errors from retrievals
using TEMPEST+ABI_3W and TEMPEST-only sensors. Among six retrieval settings, the estimated water
vapor profiles are nearly unbiased when the cloud fraction is between 0.4 and 0.6 with about 0.5 g/kg
of error standard deviation, as these amounts of clouds provide enough signals and do not entirely
obscure signals underneath. For low cloud fractions, assigning the cloud fraction from GOES-16 ABI
leads to a bias, although the standard deviation is roughly the same as if a cloud fraction of 1 is
assigned. This can be attributed to the nonlinear response of the MW radiances at 87 and 164 GHz to
cloud water content. When the assigned cloud fraction is small, the retrieval must assign all the
necessary cloud liquid water to a small cloud fraction, saturating the radiance signals and generally
causing poorer retrievals. As was seen in the synthetic retrievals, saturation will cause the cloud water
to be underestimated, which will in turn lead to an overestimation in water vapor as the OE tries to
balance all radiance terms. If the scene is truly overcast (observed cloud fraction near 1.0), there can
be no difference between assigning a cloud fraction of 1.0 as the default assumption or 1.0 as an
observed parameter, and this is reflected in the results as well.
Table 3. Column bias and standard deviation of retrieved water vapor mixing ratio in the cloudy skies
when compared to IGRA radiosonde observations. Statistics are evaluated based on all converged 51
cloudy sky cases for TEMPEST+ABI sensor configurations and 77 cloudy sky cases for using TEMPEST
and TEMPEST+ABI_3W channels.

| Sensors | Using GOES-16 cloud products | | |
|---|---|---|---|
| | No<br>set CF to 1 | Yes<br>set CF from GOES-16 | Yes<br>set CF, CH, and CP from GOES-16 |
| TEMPEST+ABI<br>(13 channels) | $0.034 \pm 1.524$ g/kg | $0.071 \pm 1.509$ g/kg | $0.488 \pm 1.816$ g/kg |
| TEMPEST+ABI_3W<br>(8 channels) | $0.007 \pm 1.440$ g/kg | $0.061 \pm 1.462$ g/kg | $0.514 \pm 1.665$ g/kg |
| TEMPEST<br>(5 channels) | $0.039 \pm 1.488$ g/kg | $0.083 \pm 1.488$ g/kg | $0.575 \pm 1.632$ g/kg |


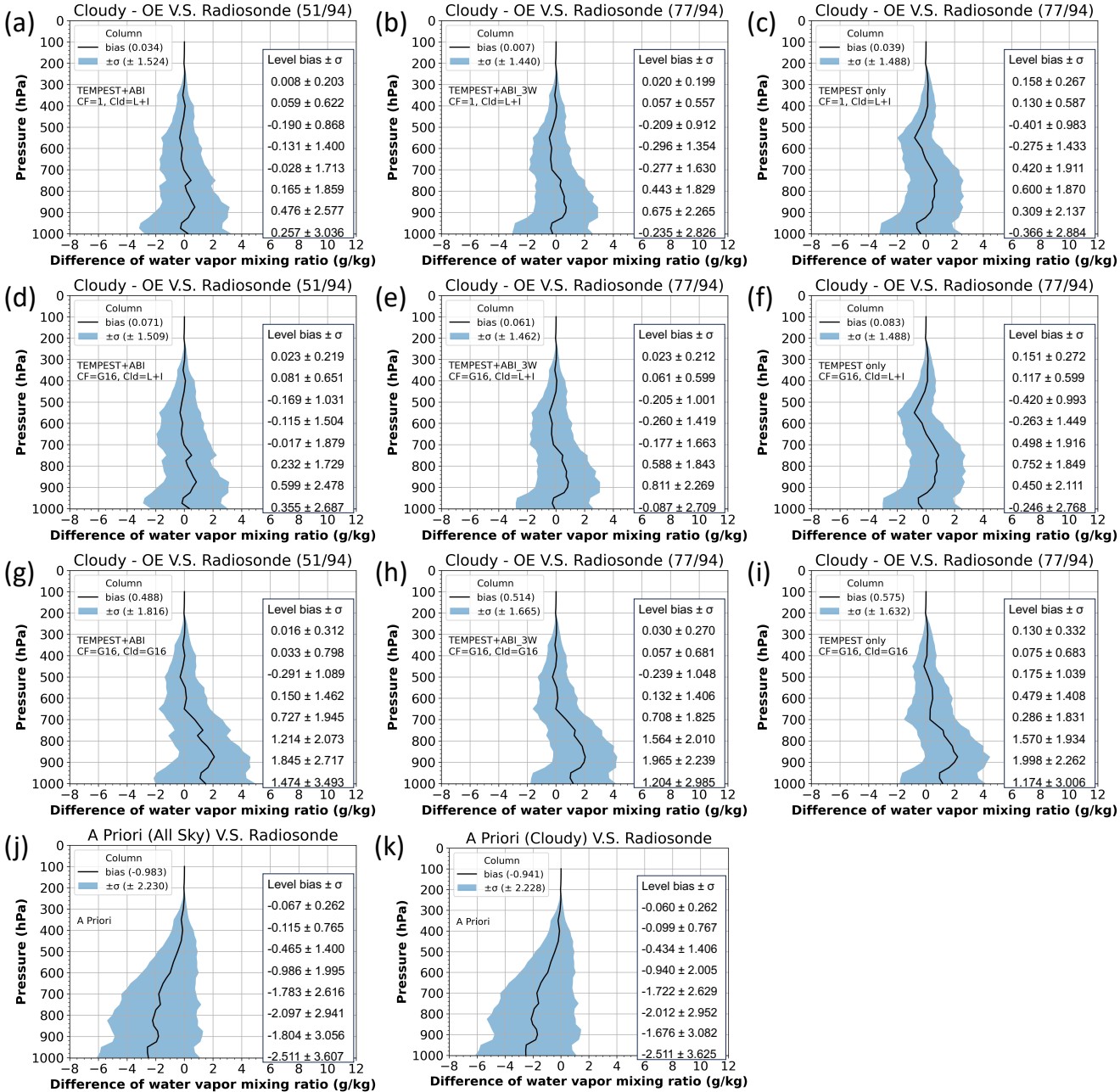

Figure 12. The water vapor mixing ratio difference between retrievals and radiosonde measurement (retrievals minus IGRA) with GOES-16 observed cloudy conditions. Retrievals use five TEMPEST channels with eight ABI bands in Figures (a), (d) and (g) and with three ABI water vapor channels in Figures (b), (e) and (h), and use only TEMPEST channels in Figures (c), (f) and (i). Figures (a) to (f) show retrievals assuming liquid and ice clouds with cloud fraction = 1 for Figures (a) to (c) and with cloud fraction from GOES-16 cloud mask for Figures (d) to (f). Retrievals in Figures (g) to (i) use cloud fraction, height, and phase from GOES-16 products to define cloud layers. Figures (a) to (c) use all-sky a priori, and Figures (d) to (i) use cloudy sky a priori. In the retrievals, the biases of the water vapor a priori information derived from all-sky conditions are shown in Figure (j) and obtained from cloudy skies are presented in Figure (k). The solid black lines are the bias value, and the blue shade regions indicate the

standard deviation (σ). The included table quantifies the retrieval performance from 300 to 1000 hPa
for every 100 hPa. The number in the parentheses means the number of all converged cases out of all
cloudy sky cases. G16 means GOES-16 products, and L+I indicates liquid and ice clouds.


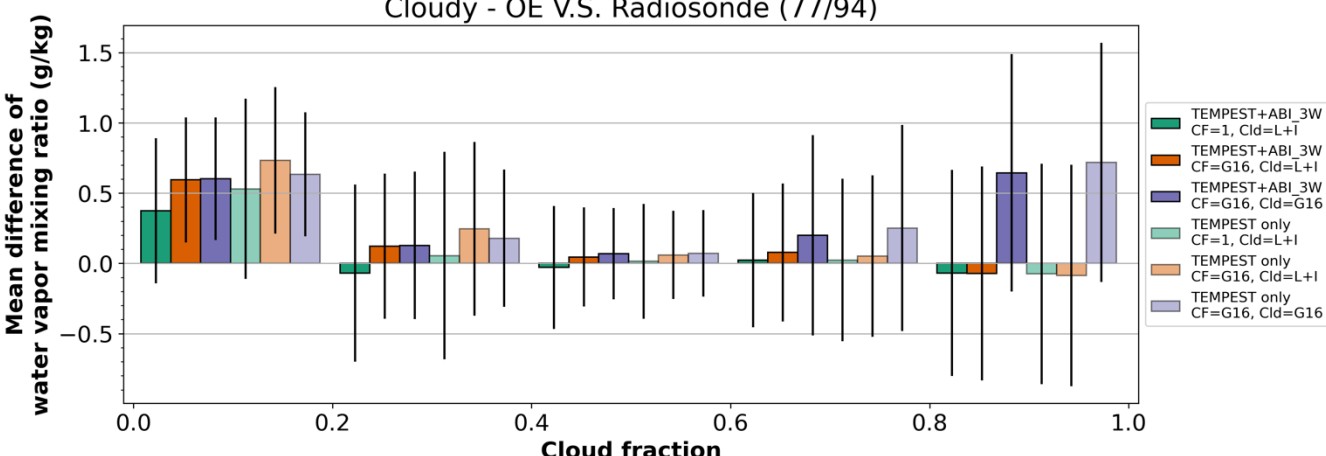

Figure 13. The mean difference between retrieved and radiosonde-observed water vapor profiles
(retrievals minus IGRA) within different GOES-16 cloud fraction intervals. Assuming both liquid and ice
clouds exist, the green bars indicate that retrievals use cloud fraction = 1, and the orange bars mean
that retrievals use only cloud fraction from GOES-16 products. The purple bars show retrievals using
cloud fraction, height, and phase from GOES-16 products. Lighter colors mean retrievals only use
TEMPEST-D, and darker colors show retrievals using both TEMPEST-D and GOES-16 ABI three water
vapor channels. Solid black lines are the range of ± standard deviation. The number in the parentheses
means the number of all converged cases among all cloudy sky cases. G16 means GOES-16 products,
and L+I indicates liquid and ice clouds.


**5.  Conclusions**

TEMPEST-D successfully demonstrated the capability of CubeSats radiometers to maintain well-
calibrated MW signals in five channels from 87 to 181 GHz over a period of almost 3 years. Although
TEMPEST-D and the TEMPEST instrument currently flying with COWVR on the International Space
Station are economical and functional, these small MW radiometers fly without an accompanying
hyperspectral IR sensor typical on operational platforms. GOES-R ABI sensors provide observations of
the Earth every 1 to 10 minutes depending on the modes, and measure 16 spectral bands from VIS to
IR with 0.5 to 2.0 km ground resolution. Given such unique ABI observations with high spatial and
temporal resolution, supplemental information from ABI enhances the ability of TEMPEST as well as
other similar CubeSats to infer the states of the atmosphere.

Along with five TEMPEST MW bands, this study presented improvements in humidity profiles that are
possible when TEMPEST retrievals are supplemented with three IR water-vapor-sounding channels and
five IR window and $CO_2$ bands available from GOES ABI. A number of positive outcomes were shown in
this paper. In the sensitivity tests comparing the combined MW/IR retrievals to MW-only capabilities,
the effective vertical resolution increases, as seen by smaller layer errors, under both clear and cloudy
conditions. The retrieved water vapor profiles were validated using independent IGRA humidity-
sounding data from 2019 to 2020. During these two years of routine TEMPEST-D operations, only 104
IGRA cases (10 cases are clear scenes, 94 under different cloudy conditions) exist. Consistent with the
sensitivity tests, the validation also showed the advantages of using GOES-16 cloud products and
additional ABI IR channels in water vapor sounding under different sky conditions.
In clear sky regions, with ABI's ability to unambiguously characterize these scenes as cloud-free,
retrievals are improved merely by forcing the scene to be cloud-free and by gaining more information
around the lower part of the atmosphere from ABI window and $CO_2$ bands. While statistics in Figs. 10
and 11 indicate that column average biases grow slightly when the ABI cloud mask is used to identify
the scene as cloud-free, the profiles themselves show clear improvement above the boundary layer.
Near the surface, retrievals are sensitive to the large biases in the prior data in these comparisons, and
it is difficult to draw conclusions. Nonetheless, adding three ABI channels slightly decreased overall
biases from 0.510 to 0.501 g/kg and biases are further reduced to 0.476 g/kg using extra five ABI
window and $CO_2$ channels with about the same error standard deviation of 1 g/kg.
Under cloudy conditions, water vapor retrievals have different degree of improvements when adding
ABI, as shown in Figs. 12 and 13, and results are generally improved when cloud fraction information is
added to the retrieval, except for very small cloud fractions where saturation in the cloudy portion of
the footprint becomes an issue. Adding cloud top and cloud phase information causes errors larger
than 0.5 g/kg. This is likely due to incorrect assumptions about the ice cloud scattering properties.
This study explored the advantages of merging TEMPEST-D, with ABI observations from GOES-16 to
improve water vapor soundings. However, ABI-like sensors, whether on the Himawari series satellites
(Bessho et al., 2016) or other platforms, cover the entire globe, providing multi-spectral, high spatial,
and high temporal observations. While we can only speculate, we assume that hyperspectral IR (Li et
al., 2022) planned for the next generation of geostationary satellites will significantly improve the
sounding capabilities in clear sky regions. This should lead to better overall retrievals in cloudy skies as
well, if one can extrapolate results from Figs. 6 and 7, which show the improvements to the passive
MW retrievals when more information is added to the retrievals. With more and more CubeSats being
launched, including COWVR and TEMPEST on Space Test Program-Houston 8
(https://podaac.jpl.nasa.gov/COWVR-TEMPEST), TROPICS (Blackwell et al., 2018;
https://tropics.ll.mit.edu/CMS/tropics), and the INvestigation of Convective UpdraftS (INCUS; van den
Heever et al., 2022; https://incus.colostate.edu) missions, these missions will all benefit from more
sounding and cloud information from ABI-like sensors or even from geostationary hyperspectral IR
sensors, enhancing the capability of CubeSats.
**Code availability**
CRTM is available through the website https://github.com/JCSDA/crtm, and MonoRTM can be assessed
by the website https://github.com/AER-RC/monoRTM.

**Data availability**

The TEMPEST-D datasets can be downloaded through the website https://tempest.colostate.edu after
registration. The GOES-16 products are archived at CLASS (https://www.avl.class.noaa.gov). The IGRA
dataset is available at https://www.ncei.noaa.gov/products/weather-balloon/integrated-global-
radiosonde-archive. The ERA5 dataset can be accessed by the website
https://www.ecmwf.int/en/forecasts/dataset/ecmwf-reanalysis-v5.

**Author contribution**

CPK and CK designed and improved the experiments. CPK is responsible for collecting and processing
data. CPK prepared the manuscript. CPK and CK discussed the results and revised the manuscript.

**Competing interests**

The contact author has declared that none of the authors has any competing interests.

**Acknowledgments**

This study was supported by NASA grant 80NM0078F0617 as part of an effort to improve water vapor
soundings from the TEMPEST CubeSat radiometer on Space Test Program-Houston 8. The authors
appreciate the reviewers' thorough comments, which greatly improved the paper.

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
