# Peer review of "Merging TEMPEST Microwave and GOES-16 Geostationary IR soundings for"

_Atmospheric Measurement Techniques, 2023_

## Author Response (AR1)

The authors appreciate anonymous referee #1's time and dedicated comments. Please find the replies to the comments below.

Anonymous Referee #1

Merging TEMPEST Microwave and GOES-16 Geostationary IR soundings for improved water vapor profiles.

This paper presents the retrieval of moisture profiles from TEMPEST with the help from 3 ABI water vapor channels with the optimal estimation method. Simulation studies as well as real data demonstrations were carried out. The simulation studies show that adding the 3 ABI water vapor channels benefits the moisture retrievals in both clear and cloudy skies. For real data demonstration, the validation with limited RAOB shows mixed results, which should be expected since real data is a lot more complicated than simulated data. The paper is well organized. Most of the sessions are well written. However, some critical points are not well explained, and some discussions are not convincing. Major revisions are recommended.

Major comments:

1. Why did authors only choose the 3 ABI water vapor channels? The window and CO2 channels, especially the 12um has low level moisture information that should be able to improve the low level moisture retrievals especially in clear sky.

R: While the window and CO2 channels, especially the 12 um channel has low-level moisture information, TEMPEST has similar low-level weighting functions. As pointed out in the reviewer's overall comments, simulation studies are always a bit idealized, and we thought this might impact the relative results. We have added the rationale for using the sounding channels to lines 135 to 137 in the revised text.

2. The authors didn't explain how the collocation between TEMPEST and ABI is done. Ideally, this should be done by considering the footprint size of both TEMPTEST and ABI. Since they have different viewing angle, the collocation will result in variable number of ABI pixels within a TEMPEST footprint. A collocation like this also allows using clear ABI pixels to help the retrievals of a partially cloudy TEMPEST footprint.

R: We have description on exactly how the footprint matching is carried out in lines 137 to 140. On the second point, the reviewer is correct that this could have been explored but was not. Such a two-step process is actually quite complicated since the clear-sky IR within the TEMPEST FOV would have to enter the retrieval as a-priori information over part of the FOV. Uncertainties, which are never as well defined as one hopes, would then be critically important.

3. Please add explanation why the auxiliary data is from ERA5, not from others. The ERA5 can actually be better used as another source for evaluation or validation of the moisture retrievals. This is less ideal than RAOB but will give you more samples to work with to gain statistically significant results. For the purpose of weather prediction, using NWP forecast instead of reanalysis data as auxiliary data is also more meaningful.

R: ERA5 was used simply because it was readily available for these studies. We now make that case. As for why it is not used for validation – we respectfully disagree with the reviewer. Until models no longer need to de-alias observations, as they currently do with microwave observations, they probably should not be used as "truth". As a personal comment, we privately do compare our results to ERA5, but we do not consider this to be validation.

4. For Table 1, please explain why you designed those configurations. For clear sky, why would you want to set CF to 1? Why don't you just perform clear sky retrievals? Why did you add CF=1 or ABI CF for clear sky samples? In table 1, for the cloudy sky, explain why you designed those configurations and what you were trying to achieve. Minor question: are you using ABI cloud mask to define clear vs cloudy sky?

R: When the pixel is known to be clear, we set cloud fraction to 0. When we use the ABI cloud mask information to define the fraction of the pixel that is cloud-covered, the cloud fraction is set by ABI. When we do not use any a priori cloud fraction information, then the cloud fraction is set to 1.0 – namely that there is a uniform cloud present, although the retrieved cloud water can be 0. We agree that the CF value of 1.0 could be confusing and we now explain that in the text in lines 199 to 203.

5. Table 2 as well, why did you design those configurations and what the differences between them? The numbers are already shown in figure 10. Does "No" mean no ABI cloud products available or you chose not to use? If "yes" means abi cloud products available, why set CF as 0? This is consistent with Figure 10 captain but not figure itself, where CF is set as G16.

R: The cases used in Table 2 and Figure 10 all have the available cloud information from GOES cloud products. However, this is not the case for every pixel. Therefore, the CF=1 setting will give readers some idea of retrieval errors when the ABI cloud mask is unavailable. We now explain that better in lines 471 to 473.

6. Explain the configurations in Table 3 as well. The numbers in Table 3 are already in Figure 12.

R: The reason for the configurations in Table 3 is the same as in Table 2. The cases used for Table 3 and Figure 12 have all cloud information, such as cloud fraction, cloud phase, and cloud top height. However, GOES cloud products do not provide cloud fraction, phase, and top height for every pixel. So, Table 3 and Figure 12 show all the results when different information is available. We now explain this better in lines 527 to 530.

7. It looks like you use CRTM for both forward simulation and Jacobian calculation for ABI channels. For CRTM, you will need to input the profiles of different cloud species water content and particle size. I assume you can get such information from ERA5 somehow, although not all information is available from ERA5. But CRTM provides Jacobians of the cloud profiles. This would be different from figure 3 which shows Jacobians of clouds are located at 900 and 300 hPa. This makes sense if you are assuming single layer of clouds at those pressure levels. So how did you handle the cloud Jacobians in the retrieval? Did you just convert the Jacobian profiles of clouds to single layer Jacobians? If so, how? Also in Figure 3, I would suggest to show moisture/cloud Jacobians as dTb/dlog(q or clouds). At 300 hPa, it doesn't make sense to perturb the moisture profile by 1 g/kg. With your current units, it is hard to say the cloud Jacobians show strong signals of clouds. In addition, it might be a good idea to show both moisture and cloud Jacobians at the cloud pressure on the figure. Currently, you are only showing cloud Jacobians at 300 and 900 hPa. Suggest to add two rows either at the top or bottom of the figure for the two cloud layers and leave the original two rows for moisture Jacobians.

R: In lines 194 to 198, we stated the settings of liquid and ice clouds clearer. Except using GOES-16 cloud products, since passive microwave have no cloud height information, in retrievals, clouds are placed in a single layer with liquid cloud top at 900 hPa and ice cloud top at 300 hPa. Therefore, liquid and ice cloud Jacobians only have values at certain layers, as in Figure 3, instead of profiles of cloud Jacobians. Figure 3 shows Jacobians from the TOA to the surface for more intuitive purposes. The Jacobians are derived by perturbing 1 % of humidity or cloud water path, so it isn't necessary to perturb by 1 g/kg water vapor or 1 $g/m^2$ cloud water path. The patterns of Figure 3 are about the same by using dTb/d(q or clouds) or dTb/dlog(q or clouds). The reason for showing Figure 3 is to present the humidity and cloud sensitivity along the height for TEMPEST and ABI water vapor channels. We added Jacobians in the clear sky in Figure 3 for comparisons.

8. Discussion of figure 10 from line 478 to 494 is confusing. These are clear sky profiles (assuming you use ABI cloud mask). Why would you set CF=1 for figure 10 a and b? Line 488-490, what do you mean by "forced to be cloud free"? These are categorized as clear profiles based on ABI cloud mask (I am assuming). Unless you don't trust ABI cloud mask, they should be clear profiles and no clouds are needed in the retrieval. Line 490- 494, what did you use as a priori? Figure 2 shows clear profile below 900 hpa is wetter than cloudy, not consistent with figure 10 e and f. Again, this is clear profiles, why would you suggest to add clouds in the retrievals? What is the bias from ERA5 information in Figure 10f under clear sky assumptions? As compared to truth? What do you mean bias is "even larger"? Since bias could be positive or negative, suggest using more dry biased or wetter biased to indicate the bias change direction. Did you use different a priori for different samples, as shown in Figure 10 e and f? Figure 2 indicates you use the same climatology a priori for all samples.

R: Figure 2 shows the water vapor monthly mean in May 2020 for clear and cloudy skies, and Figure 10 shows the retrieval and climatology (a priori) biases for 4 scenarios, as shown in Table 1. It is intended only to demonstrate the impact of the data sources. The purpose behind each scenario is now explained on lines 198 to 203 and 471 to 473.

9. Discussion of figure 12 is also not very convincing. Line 510-511, improvement of moisture retrievals? Which one compared with which one? If between TEMPEST+ABI and TEMPEST only, yes, there is some improvement, not a lot and not consistently of all levels above 800 hPa. If between a and c or b and d, I see even smaller changes. And I don't know which one is expected to be improved because you never explained the differences and purposes of these configurations. Line 517, how do you know it is correct cloud fraction? Line 519 to 523, how do you know the cloud fraction is too small? The cloud fraction is small. But that doesn't necessarily mean it is too small. Line 530-531, not all levels above 700 hPa.

R: We have tried to be more precise in the discussion of figure 12, referring quantitatively to specific improvements in lines 533 to 546. We modified the expression about cloud fraction as reviewer's comments in lines 551 and 554, and also described the detail comparisons in lines 563 to 567 instead of only expressing "levels above 700 hPa".

Specific comments:

1. Line 70, IASI naming wrong

R: Thanks for pointing this out. We modified it to the Infrared Atmospheric Sounding Interferometer (IASI) in lines 70 to 71.

2. Lines 83-86, NUCAPS has been extended to Metop as well

R: NUCAPS includes Joint Polar Satellite System and Metop series satellites. However, we mentioned sensors and not satellites in lines 83 to 86. We added IASI in this context to correctly indicate the ability of NUCAPS in line 85.

3. Lines 152-155, these absorption channels are sensitive to both moisture and temperature. We can't say one is more sensitive than the other. You didn't need to worry about temperature because you are assuming temperature is known from ERA5.

R: We removed the sensitivity statement in previous lines 152 to 155.

4. Lines 179-182, please clarify if you are using the same xa for both clear and cloudy profiles.

R: We made it clear to reflect that retrievals use clear (cloudy) sky $x_a$ when GOES cloud mask indicates the pixel is clear (cloudy) or use whole sky $x_a$ when GOES cloud mask is not used in retrievals in lines 185 to 187.

5. Line 192, is available from GOES-16

R: We modified line 192 as in revised manuscript in line 198.

6. Line 207, please specify these NEDT at what BT

R: We added that these ABI NEDT values are evaluated at 300 K in line 217.

7. Line 300, This figure will be better shown if you zoom into a few TEMPEST footprints and show how ABI is collocated with it. One panel for BT and the other for cloud mask.

R: Figure 4 shows the collocated TEMPEST-D and GOES-16 ABI observations. Since ABI observation has been averaged to match the TEMPEST-D spatial resolution and we also want to show the observation area of GOES-16 ABI, the global view of collocated observations would be better.

8. Line 321, this is just one sample. You can only calculate the mean bias and STD of whole profile. There is no such thing as mid tropospheric biases and STD. Same is true for line 324-325.

R: We modified these lines to mention only bases in line 334.

9. Line 338, bias +/- standard deviations of the whole profile.

R: We specified the whole profile in line 351.

10. Line 344, why did you only choose 1000 not all available samples. More samples are more statistically meaningful. Same as line 364.

R: We explained the reason selecting 1000 samples in lines 358 to 362. Since in the observations on 2020/05/27 all clear sky pixels are about 1200 samples and about 8400 cases are cloudy pixels according to the GOES-16 cloud mask, we randomly selected 1000 samples over clear and cloudy cases to have fair comparisons between them. The statistics are about the same, no matter how we randomly selected the 1000 clear or cloudy samples.

11. Line 367, it is not true that ABI has less sensitivity in lower atmosphere. It all depends on the moisture and cloud condition. ABI has more sensitivity to upper troposphere in wet condition and less sensitivity to lower troposphere in dry condition. Clouds overall reduce moisture sensitivity. But it's hard to see that from Figure 3 because you didn't show Jacobians in clear sky for comparison.

R: We added clear-sky Jacobians for comparisons. Under clear or cloudy skies, TEMPEST 87 and 164 GHz channels all have stronger sensitivity in the lower atmosphere than ABI 3 water vapor channels.

12. Line 375-378, authors hinted that the clear profiles are drier than cloudy, which is true for most levels but not below 900 hPa according to Figure 2. Using same a priori for both clear and cloudy profiles would mean the first guess is dry bias below 900 hPa. That is probably the reason why Figure 6 sees more bias below 900 hPa.

R: Yes, lines now in 392 to 397 explain that the larger water vapor retrieval biases are due to a priori information.

13. Line 390, why only choose 8000 samples not all samples?

R: The number of cloudy pixels is about 8400. Randomly selected 8000 samples are about all the cases, and the statistics are the same over 8400 or 8000 cases. We added total number of cloudy pixels in lines 358 to 362 and line 409.

14. Line 404-406, did you mean IR has less information about ice water content than MW?

R: No, we did not imply that IR has less ability to retrieve ice water content than MW. We removed the lines to avoid the implication.

15. Figure 10 e and f, why is clear sky drier than cloudy in lower troposphere while figure 3 shows the opposite. Aren't you using the same a priori for all retrievals?

R: Figures 10(e) and 10(f) show biases between a priori and radiosonde for the 10 clear sky cases from 2019 to 2020, and the a priori is not from a specific month, such as Figure 2, but from each collocated month. That is why humidity profiles in Figures 10(e) and 10(f) are dryer for clear sky a priori.

16. Line 482-482, same argument should be applicable to simulation studies.

R: Now in lines 501 to 505, Figures 6 and 10 show the consistent results that retrieved clear-sky water vapor profiles in the 1000 to 800 hPa layer were not improved after adding 3 ABI channels.

17. line 595, improvements between with and without 3 ABI water vapor channels? Line 596, what do you mean by "when ABI is used to identify the scene as cloud free"?

R: The improvements mentioned in previous line 595 and now in line 631 are that clouds will not be retrieved in ABI-identified clear skies. In previous line 596 and now in line 632, the statement "when ABI is used to identify the scene as cloud-free" refers to the usage of the ABI cloud mask in retrievals and we added cloud mask in context to avoid confusing.

The authors appreciate anonymous referee #2's time and suggestions. Please find the replies to the comments below.

Anonymous Referee #2

The manuscript demonstrates the synergistic use of cubesat sounding technology with existing instruments from the program of record to obtain science quality water vapor soundings. This paper makes a timely contribution to the decision making process concerned with future sensor acquisition strategies.

I recommend the paper is accepted, pending minor revisions as indicated below.

1) Using ERA5 for ancillary data comes with some error. The authors should try to add a study to evaluate the impact of the temporal and spatial mismatch on product performance. This can be particularly important when it comes to cloudy scenes.

R: In lines 155 to 158, we added expression about the possible retrieval errors from the ERA5 ancillary data.

2) The authors could draw some expected conclusion from the use of hyperspectral IR geostationary instruments. Of particular importance would be the use of more water vapor channels and the use of hyperspectral IR retrieved temperature and ancillary products in replacement of ERA5.

R: The geostationary hyperspectral IR instrument is a powerful sensor, and the potential use of the sensor with CubeSat satellites is appealing. We added the statement in the conclusion in lines 647 to 651 and lines 656 and 657.

---

## Author Response (AR2)

The authors appreciate the reviewer's detail suggestions. The replies to the comments are addressed below in green color.

Major comments:

1. It is true that the ABI window and CO2 bands may have overlapped information with TEMPEST window channels. But that doesn't mean it is redundant. Just like hyperspectral IR sounders, more channels mean more information content and more useful for retrievals.

R: Other than the ABI window (channels 13 to 15) and ABI CO2 bands (channel 16), we also added an ABI dirty window band (channel 11) in the retrievals along with ABI water vapor bands (channels 8 to 10) in Figures 1, 3, 5 to 7, and 10 to 13, Tables 1 to 3, and in the corresponding context. Using ABI channels 8 to 11 and 13 to 16 helps TEMPEST to retrieve water vapor in clear skies, compared with retrievals using TEMPEST and ABI water vapor channels 8 to 10. However, in cloudy skies, since ABI channels 11 and 13 to 16 have less sensitivity below clouds, using these extra channels does not improve humidity retrievals.

2. About the collocation, it is still not clear how it is done. Both ABI and TEMPEST footprint sizes change with zenith angle. So the number of ABI pixels fall into a TEMPEST footprint should be variable. How did you find the ABI pixels falling into a TEMPEST footprint? And how did you calculate the cloud fraction for TEMPEST footprints using ABI cloud information?

R: The ABI and TEMPEST data used in the study are all geolocated. The collocation is performed by using the nearest matched latitude and longitude information along with the data, so several ABI pixels will be matched with a single TEMPEST pixel. The collocated ABI pixels, such as brightness temperature, cloud phase, cloud height, and cloud fraction, are averaged to match TEMPEST pixels, as mentioned in lines 143 to 146.

3. It may not be appropriate to call it a validation, but comparison of sounding retrievals to reanalysis or analysis is a common practice. However, the bigger issue is that using ERA5 may overestimate the retrieval results. ERA5 is arguably more accurate than other datasets. Using it to provide temperature profile information is equivalent to assume an

accurate temperature profile is known. So the moisture retrieval quality might be overestimated. Please add discussion on this.

R: In Section 4.1, since data from ERA5 is assumed to be the true atmospheric conditions, the sensitivity tests can be performed by using simulated observations from ERA5 to evaluate the retrieval accuracy. In Section 4.2, the humidity retrievals are evaluated by comparing independent water vapor data from radiosonde measurements, so we think it is appropriate to call the Section 4.2 as Independent Validation. The issues of the appropriateness of ERA5 in retrievals when comparing with in situ humidity measurements are described in lines 161 to 164.

4. The Jacobians of Figure 3. It is clear that (c) is water vapor Jacobian (dTb/dq). But what is it in (d) at 300 and 900 hPa? It is water vapor Jacobian or cloud Jacobian (cloud liquid/ice water path)? It can't be both. That is why I originally suggested to add two rows for clouds. If it is water vapor Jacobian, what do smaller Jacobians at 300 and 900 hPa mean?

R: As mentioned in the caption of Figure 3, the Jacobians at 300 and 900 hPa are for water vapor (dTb/dq) for Figure 3 (c) and for ice and liquid clouds (dTb/dCWP) for Figure 3 (d). The reason for inserting cloud Jacobian between water vapor Jacobian is to provide an overview of the microwave and IR vertical sensitivity along the height. To avoid confusion, we removed the water vapor Jacobians at 300 and 900 hPa for Figure 3 (c) and kept ice and liquid cloud Jacobians at 300 and 900 hPa for Figure 3 (d). In this way, the influence of clouds can be seen in Figure 3 (d) in comparison with Figure 3(c).

5. The above comment leads to two more questions: a) what exactly are the variables that you are retrieving? Is it moisture profile in clear sky and moisture profile plus CIWP/CLWP in cloudy? If so, are you assuming all other information is known, such as temperature, cloud cover, etc? This needs to be explicitly noted. b) what inputs are used for ABI cloudy radiance simulation using CRTM? CRTM takes cloud profiles of water/ice content and particle sizes. You have detailed explanation of how microwave clouds are handled in lines 259-276, but not much for ABI.

R: As described in lines 195 to 199, the retrieval variables are only water vapor mixing ratio in the clear sky and in the cloudy sky are liquid/ice cloud water path as well as water vapor mixing ratio. Meanwhile, the cloud macrophysical properties are mentioned in lines 200 to 209 and in Table 1, basically saying that clouds are configured depending on the different degree of usage of GOES-16 cloud products. The cloud vertical structure is in single layer for both liquid and ice cloud with cloud top height at 300 hPa for ice clouds and at 900 hPa for liquid clouds. The cloud settings for CRTM are in lines 260 to 264, using default CRTM cloud optical properties with 12 and 30 μm effective radius for liquid and ice clouds, respectively.

6. The VIS/IR has different sensitivities to clouds than microwave. So the cloud information from ABI might not be the best representative of that from microwave. Discussion on the impact of using ABI cloud information for microwave sounding retrievals is needed.

R: This could be an uncertainty source, so in lines 613 to 616, we added discussion about possible inappropriateness of using ABI derived cloud properties in MW retrievals.

7. Discussion of Figures 10-13 is not convincing. When evaluating profile retrievals, we need to look at both bias and STD to have a good understanding of the quality. If just using one parameter, RMSE which includes both bias and STD, should work. See specific comments below.

R: In Figures 10 to 13, we showed the biases and standard deviations of retrievals for different levels and for whole vertical column under different sensor configurations using different degree of cloud information from ABI cloud products. In the discussion, not only biases but standard deviations are described to compare among different sets of retrievals. We think these Figures show the comprehensive outcomes from different aspects.

Specific comments:

1. Line 231, using the reduced prescribed levels? This was not defined as an assumption.

R: The reduced prescribed levels are the levels used to retrieve water vapor, as mentioned in lines 197 to 199. To avoid confusion, we modified it to "the previous prescribed retrieval levels" in line 232.

2. Figure 1, add the information of the unit.

R: We added unit of the colorbar for in line 247.

3. Line 365-366, it is not a true statement that ABI has little influence on the low level moisture. It is because you didn't use window and CO2 bands.

R: We added ABI window and $CO_2$ bands in the retrievals and these channels help to reduce retrieval errors in clear skies. The influences of using the ABI window and $CO_2$ bands are discussed in the manuscript.

4. Figure 5, is this just one profile or average of multiple profiles? Either way, make it clear in the caption.

R: In line 369, we used "Two selected cases of retrieved water vapor profiles" to avoid confusion.

5. Line 386-393, suggest: Based on ABI/GOES-16 cloud mask, there are about 1200 clear sky and 8400 cloudy pixels successfully collocated with TEMPEST on May 27 2020. Randomly selecting 1000 samples from both clear and cloudy pixels allows fair statistical comparisons between clear and cloudy regions. The statistics are found independent of how the 1000 samples are randomly selected.

R: We have modified the description according to the reviewer's suggestions in lines 384 to 388.

6. Line 404, why did you use all-sky a priori not clear sky a priori?

R: As Figures 6 and 7 in Section 4.1, we used all sky a priori in the sensitivity tests to compare retrievals. In this way, we can discuss possible error sources as well as compare retrievals in clear or cloudy skies. In the validations (Section 4.2), we used a priori from clear, cloudy and all skies depending on different degree of cloud information from ABI.

7. Figure 10, why the clear cases are better retrieved when assuming they are cloudy. If you calculate RMSE, you can see that c has larger RMSE than a, and d has larger RMSE than b.

R: This is discussed in lines 554 to 562, saying that if cloud retrievals are allowed water vapor retrievals are underestimated as parts of the water vapor signal are attributed to be clouds, otherwise, water vapor retrievals are overestimated. Or it is possible that the small number of validation cases is not representative.

8. Lines 589-591, it is hard to consider those are improvements because STD is increasing.

R: As in lines 573 to 616, although some settings have clearly improvements over biases and STDs, the other settings have improvement at certain levels. For example, while column STDs increase in Figures 12(g) to 12(i), the column biases are decreased comparing retrievals with using TEMPEST-only and adding ABI sensors. The column STDs are increased about 2 %, but column biases are reduced by about 11 %. We do think it is improvements according to larger reduction in column biases than STD increments.

9. Line 591, suggest to start a new paragraph starting at "The water vapor retrieval errors...."

R: We have made it to the new paragraph as in lines 573 to 616.

10. Lines 598-601, I don't agree with the discussion here. Cloud fraction being small does not mean cloud fraction has a negative bias (too small), which can cause saturation of

cloud content. This actually leads to the major comment of collocation. How did you calculate the cloud fraction of a TEMPEST FOV from ABI cloud information?

R: The collocated cloud fraction is calculated by searching the nearest geolocated ABI cloud mask to the geolocated TEMPEST pixels. The matched ABI cloud masks are average to represent the larger TEMPEST pixels. Lines 627 to 631 are saying that when cloud fraction is small the retrieved cloud water path will be larger than cloud water path retrieved in high cloud fraction situations and brightness temperature is saturated when cloud water path is large, so combining these two effects causes poor retrievals in the low cloud fraction environment.

11. Lines 608-612, I don't think bias reduction with STD increasing is significant improvement.

R: We have modified the context to only "improvement" in line 604 and to "water vapor retrievals have different degree of improvements" in line 704.

12. Line 630, where is the overfitting?

R: In line 610, we modified the statement to "some overfitting appears to be taking place between 700 and 1000 hPa" to indicate where the overfitting is.